

# Modelling deep-water formation in the North-West Mediterranean Sea with a new air-sea coupled model: sensitivity to turbulent flux parameterizations

Léo Seyfried[1], Patrick Marsaleix[1], Evelyne Richard[1], and Claude Estournel[1]

[1]Université de Toulouse, CNRS, UPS, Laboratoire d'Aérologie, Toulouse, France

*Correspondence to:* Léo Seyfried (leo.seyfried@aero.obs-mip.fr)

**Abstract.** In the north western Mediterranean, the strong, dry, cold winds, the Tramontane and Mistral, produce intense heat and moisture exchange at the interface between the ocean and the atmosphere leading to the formation of deep dense waters, a process that occurs only in certain regions of the world. The purpose of this study is to demonstrate the ability of a new coupled ocean-atmosphere modelling system based on MESONH-SURFEX-SYMPHONIE to simulate a deep-water formation event in real conditions. The study focuses on summer 2012 to spring 2013, a favourable period that is well documented by previous studies and for which many observations are available. Model results are assessed through detailed comparisons with different observation data sets, including measurements from buoys, moorings and floats. The good overall agreement between observations and model results shows that the new coupled system satisfactorily simulates the formation of deep dense water and can be used with confidence to study ocean-atmosphere coupling in the north-western Mediterranean. In addition, to evaluate the uncertainty associated with the representation of turbulent fluxes in strong wind conditions, several simulations were carried out based on different parameterizations of the flux bulk formulas. The results point out that the choice of turbulent flux parameterization strongly influences the simulation of the deep water convection and can modify the calculated volume of the deep water formed by up to one order of magnitude.





# 1   Introduction

The North-West Mediterranean Sea (NWMS) is one of the few regions in the world where the deep open ocean convection process is regularly observed (Marshall and Schott, 1999). The strong, dry, cold winds, the Tramontane (north-westerly) and the Mistral (northerly), play a major role in this process. These winds induce intense exchanges between the atmosphere and the sea (Flamant, 2003; Hauser et al., 2003), with a marked loss of surface buoyancy (Schott and Leaman, 1991).The oceanic deep convection can be separated into three phases. In autumn, a cyclonic gyre, bounded to the north by the Northern Current (NC) (Millot, 1999) and to the south by the North Balearic Front (NBF) (Millot and Taupier-Letage, 2005), isolates a weakly stratified water mass whose stratification is progressively eroded by northerly winds (preconditioning phase). In winter, in some years, vertical mixing induced by strong winds leads to the formation of a vertically homogeneous water mass (convective phase) identifiable by its temperature and salinity properties, and generally referred to as new Western Mediterranean Deep Water (nWMDW). After the convective phase, the mixed zone undergoes a restratification while the nWMDW is spread throughout the basin by the general circulation (Schott et al., 1996) and submesoscale eddies (Testor and Gascard, 2006) (restratification/spreading phase).

Two components of the MISTRALS programme (http://www.mistrals-home.org/) focused on the study of deep convection in the NWMS and made a major effort in collecting observations during 2012 and 2013. The first component, HyMeX (Drobinski et al., 2013), studied the atmosphere-land-ocean coupled system. In this context, two Special Observation Periods (SOPs) were organized; SOP1 in autumn 2012, during the preconditioning phase (Ducrocq et al., 2013), and SOP2 in winter 2013, during the convection phase (Estournel et al., 2016b). The second component, MERMEX (Sempéré et al., 2010) studied the impact of oceanic deep convection on the nutrient balance and the planktonic ecosystem. Three oceanographic cruises were organized by MERMEX: DOWEX in autumn 2012, DEWEX leg 1 in February 2013 and DEWEX leg 2 in April 2013. Combining all these different surveys provided a good description of the spatial distribution and temporal evolution of ocean stratification. Furthermore, the volume of dense water formed by deep convection could be evaluated thanks to the optimal interpolation of the many Conductivity-Temperature-Density (CTD) profiles available (Waldman et al., 2016b). This series of campaigns provided a unique opportunity to test the ability of models to simulate the different phases of the dense water formation process in the western Mediterranean.

Several modelling studies of the formation of deep water in the NWMS, have been carried out over different periods and with different models [e.g. Herrmann et al. (2008); Herrmann and Somot (2008); Herrmann et al. (2010); Léger et al. (2016); Estournel et al. (2016a); Waldman et al. (2016a)]. Their results show high sensitivity to the initial stratification of the ocean model and to the accuracy of the atmospheric forcing. An attempt to progress on these two issues was proposed by the HYMEX and MERMEX groups. Concerning the first point, since 2010, oceanographic cruises have been organized in the NWMS each summer by the Mediterranean Ocean Observing System for the Environment (MOOSE; www.moose-network.fr/). These cruises provide a sample of the different water masses of the NWMS based on about 70 CTD profiles. In particular, the observations collected in summer 2012 have been assimilated to provide a more realistic initial state for ocean models (Estournel et al., 2016a; Léger et al., 2016) which has been shown to be crucial for the simulation of the winter convection 6 months later.



Regarding the atmospheric forcing, besides the questions related to its space and time resolution, there is still significant uncertainty as to the choice of a relevant parameterization to compute the turbulent fluxes for strong wind conditions such as the Mistral and Tramontane (Estournel et al., 2016b).

These issues, among others, have motivated the recent development of a new modelling platform providing better numerical tools to address the scientific and technical questions related to ocean-wave-atmosphere coupling (Voldoire et al., 2017). In particular, this new platform greatly facilitates the coupling of the different atmospheric and ocean models developed in the French community. The benefit of using a fully coupled system to study air-sea interactions in the NWMS was already illustrated in previous studies (based upon different air-sea coupling platforms) (Lebeaupin Brossier and Drobinski, 2009; Small et al., 2012; Renault et al., 2012).

A first objective of the present study is to show the capacity of the new coupled regional ocean-atmosphere system MESONH-SURFEX-SYMPHONIE to reproduce the formation of deep dense waters during the winter of 2013, in the NWMS. A second objective is to study the sensitivity of the simulations to the parameterization of the turbulent surface fluxes by testing three different parameterizations (Fairall et al., 2003; Andreas et al., 2015; Moon et al., 2007), all based on bulk formulas.

This paper is organized as follows. Section 2 describes the coupled modelling system, and the three parameterizations mentioned above that were used for the computation of turbulent fluxes. Section 3 presents the different observation data sets used to evaluate the model results while section 4 details the setup of the numerical experiments. Results are analysed and discussed in section 5. Some conclusions and perspectives are presented in section 6.

## 2 Model description

### 2.1 Air-Sea Coupled model

The modelling system is based on the non-hydrostatic atmospheric model MESONH [Lafore et al. (1997); http://mesonh.aero. obs-mip.fr/mesonh52/] developed at the Laboratoire d'Aérologie (LA) and Centre National de Recherches Météorologiques (CNRM), the multi-surface model SURFEX [Masson et al. (2012); http://www.cnrm-game-meteo.fr/surfex/)] developed at CNRM, and the Boussinesq hydrostatic ocean model SYMPHONIE [Marsaleix et al. (2008, 2009, 2012); http://sirocco. omp.obs-mip.fr/ocean_models/S-model] developed at LA. These models are coupled through the SURFEX-OASIS interface (Voldoire et al., 2017) using the OASIS3-MCT coupler (Valcke et al., 2015). The main characteristics of the atmospheric and oceanic model configurations are listed in Table 1. The oceanic model uses a horizontal resolution of 1 km. Given the value of the Rossby radius (5-10 km in the NWMS), 1-km grid spacing appears to constitute a reasonable compromise between the computing cost and the necessary resolution. In the vertical, 40 generalized sigma levels are used; 10 of them in the first hundred meters (above the abyssal plain). The resolution just below the sea surface is 1.5 m. The atmospheric model is run with a 10-km horizontal grid spacing and 52 terrain-following vertical levels ranging from 15 m to 15000 m. With such resolutions, both atmospheric and oceanic convection must be parameterized. In the case of the ocean, as described in Estournel et al. (2016a), the heat and water fluxes are re-distributed over the whole mixed layer. This parameterization is consistent with the nearly-linear vertical variation of the buoyancy flux in the convective layer (Deardorff et al., 1969). Regarding the atmosphere,





shallow and deep convection are parameterized with mass-flux schemes according to Bechtold et al. (2001) and Pergaud et al. (2009), respectively.

In the coupled system, the surface fluxes are computed by SURFEX on the atmospheric model grid. They are sent to the ocean model by the OASIS3-MCT coupler, which also performs their interpolation on the ocean model grid. Conversely, the ocean model computes the sea surface temperature and sends it to SURFEX using the OASIS-MCT coupler, which takes care of its interpolation on the atmospheric model grid. The coupling frequency is set to 10 minutes and the interpolation between the two model grids is bilinear.

The computational domains used for this study are presented in Fig. 1. The atmospheric grid covers the whole western Mediterranean basin while only part of it is covered by the ocean grid (blue area in Fig. 1) since the Alboran Sea and part of the Tyrrhenian Sea were excluded to avoid straits issues. Outside the ocean grid, the air-sea fluxes are computed using the sea surface temperature provided by the OSTIA data base (Donlon et al., 2012), the horizontal resolution of which is about 6 km.

## 2.2 Turbulent flux parameterizations

The turbulent fluxes at the air/sea interface are calculated from bulk-type parameterizations based on the Monin-Obukhov similarity theory (Foken, 2006). These parameterizations compute the turbulent fluxes as a function of the vertical gradient of the mean fields at the air-sea interface (Eqs. 1, 2 and 3)

$$|\boldsymbol{\tau}| = \rho_a C_d U^2 \tag{1}$$

$$H = -\rho_a C_p C_h (\theta - SST) U \tag{2}$$

$$LE = -\rho_a L_e C_e (q - q_{sat}(SST)) U \tag{3}$$

*where $\tau$ is the momentum flux, $H$ the sensible heat flux, $LE$ the latent heat flux; $\rho_a$ the air surface density, $C_p$ the specific heat of air , $L_e$ the latent heat of vaporization , $U$ the wind speed at 10 m, $\theta$ the air potential temperature at 10 m, $SST$ the sea surface temperature, $q$ the specific humidity at 10 m, $q_{sat}(SST)$ the specific humidity at saturation for air in contact with the sea surface, and $C_d$, $C_h$ and $C_e$ the transfer coefficients for momentum, heat and moisture.*

In this study three different parameterizations are tested :

- The COARE3.0 parameterization (Fairall et al., 2003) is one of the most widely used in the modelling community but was established for winds weaker than 20 m s$^{-1}$.

- The ANDREAS parameterization (Andreas et al., 2015) is a more physically-based approach distinguishing the fluxes at the air-sea interface (computed with COARE2.6 (Fairall et al., 1996)) on the one hand, and the sea spray fluxes (calculated with a fast microphysical algorithm) on the other hand. This parameterization was established with observations for wind speeds up to almost 25 m s$^{-1}$ .

- The MOON parameterization (Moon et al., 2007) proposes an alternative parameterization of the momentum transfer coefficient ($C_d$) which was developed and validated for strong winds and cyclones from wave–wind coupled simulations



and applied to the Geophysical Fluid Dynamics Laboratory coupled hurricane–ocean prediction model (Kurihara et al., 1998). It is worth noting that the new formulation of the momentum transfer coefficient also impacts the turbulent heat fluxes since, in MOON, $C_h$ and $C_e$ are given as a functions of $C_d$.

COARE3.0 is the standard parameterization used in SURFEX, while ANDREAS and MOON were implemented in SURFEX
specifically for the purposes of the study.

To illustrate the discrepancies between these parameterizations, Fig. 2 shows the turbulent fluxes plotted as a function of the 10 m wind speed and computed with the different parameterizations. Although not used further in the following, the results of COARE2.6 have been added for completeness. They also allow the impact of the sea spray in ANDREAS to be distinguished. The computations were carried out under unstable conditions typical of Mistral and Tramontane events : the
10 m air temperature fixed at 10 °C, the sea surface temperature at 15 °C and the 10 m relative humidity at 70 %. In these conditions, COARE3.0 and ANDREAS produced very similar momentum fluxes while the MOON momentum flux was found to be slightly weaker but only for the strongest winds (>15 m s$^{-1}$). For the sensible heat flux, COARE3.0 and ANDREAS provided close results up to a wind speed 8 m s$^{-1}$ but, for higher wind speeds, the impact of the sea spray considerably enhanced the ANDREAS sensible heat flux. Among the three parameterizations, MOON was the one that produced the strongest sensible
heat flux over a wide range of wind speeds (from 6 m s$^{-1}$ to 23 m s$^{-1}$). The latent heat fluxes presented the same hierarchy in intensity as the sensible heat fluxes. However, the departure between COARE3.0 and ANDREAS only occurred from wind speeds greater than 16 m s$^{-1}$ (as compared to 8 m s$^{-1}$ for the sensible heat flux). To summarize, the three parameterizations gave significantly different results, particularly MOON, which produced the largest turbulent heat fluxes in a wide range of wind speeds. It should also be noted that the departure between the parameterizations may become noticeable from wind speeds
as low as 8 m s$^{-1}$ and can dramatically increase with the wind speed, especially for the heat fluxes.

## 3   Available Observations

Several in situ datasets, collected by the MOOSE, HyMeX and MerMeX programmes, are used in the present study and are briefly described below. More details are available in Estournel et al. (2016b). First, the Lion meteorological buoy provides hourly measurements of the air temperature and humidity at 2 m above the sea surface, the wind speed at 10 m, and the sea
surface temperature. A mooring named LION, part of the MOOSE network, and located only 5 km away from the Lion buoy is also available. It provides vertical profiles of temperature and salinity, sampled over 21 levels for temperature and 15 for salinity, between 150 and 2300 m deep (Houpert et al., 2016).

In addition, several oceanographic cruises were carried out in the NWMS during 2012-2013 (Waldman et al., 2016b). In August 2012, the annual cruise organized by MOOSE for the monitoring of the NWMS, provided 69 vertical surface-to-
bottom profiles of temperature and salinity. The same network of CTD stations was deployed in February 2013 and April 2013, during the DEWEX leg 1 and leg 2 cruises. A smaller network was implemented in autumn 2012 during the DOWEX cruise, with only 41 profiles. Furthermore, 399 temperature and salinity profiles measured by 14 Argo floats, 4 of them specifically





deployed by HyMeX with a daily cycle, are also taken into consideration in this study. The above-mentioned cruises allowed the spatial distribution and temporal evolution of ocean stratification to be well described.

Ocean stratification is commonly assessed using the Stratification Index (*SI* Eq. 4).

$$SI(Z) = \int\limits_{Z}^{0} (\rho(Z) - \rho(z)) dz \tag{4}$$

*where $\rho$ is the potential density and $Z$ the reference level. SI is expressed in* $\mathrm{kg\,m^{-2}}$.

Note : that $SI(Z) = 0$, means that the water column is mixed at least to depth Z or, in other words, that the mixed layer depth is greater than or equal to Z.

Figures 3 (a-d) show the Stratification Index *SI* at 1000 m (Eq. 4) for each seasonal cruise and Fig. 3 e its time evolution from summer 2012 to spring 2013. In summer (Fig. 3 a), the less stratified water ($SI(1000\ m) < 80\ \mathrm{kg\,m^{-2}}$) is confined to the north of the deep basin (above 42 °N) and corresponds to the Modified Atlantic Water (MAW), while the more stratified water ($SI(1000\ m) > 120\ \mathrm{kg\,m^{-2}}$) in the south corresponds to the Atlantic Water (AW). These two water masses are separated by the North Balearic Front. In autumn (Fig. 3 b) the northward displacement of this front brings AW toward the north of the basin, whereas the oceanic stratification increases in the south of the domain. In winter (Fig. 3 c), the oceanic deep convection is revealed by many profiles with $SI(1000\ m) < 1\ \mathrm{kg\,m^{-2}}$, meaning that these profiles are mixed down to 1000 m depth. When the vertical mixing reaches the seafloor, it can then lead to the formation of dense deep water. Finally, in spring (Fig. 3 d), restratification occurs with an increase of stratification in the convective area ($SI(1000\ m) > 10\ \mathrm{kg\,m^{-2}}$).

Combining all these available observations, Waldman et al. (2016a) estimated the volume of the dense deep water formed at $4.5(\pm 1.1) \times 10^{13}\ \mathrm{m^3}$ (for density greater than 29.11 $\mathrm{kg\,m^{-3}}$) between summer 2012 (MOOSE 2012 cruise) and spring 2013 (DEWEX leg 2 cruise) in the convective area defined between [2.5 °E; 9 °E] and [40 °N; 44 °N] and limited by the 2000 m bathymetry contour (red line in Figs 3 a-d)

## 4 Numerical Experiments

To model this well documented event of dense water formation in the NWMS, three coupled simulations are reported here. They differ only in their turbulent flux formulation and are named COARE, ANDREAS, and MOON according to their parameterizations. These simulations start on 16 August 2012 with a realistic initial oceanic state and end on 1 May 2013. Initial and boundary conditions of the atmospheric model are provided by the 6-hour analyses produced by the European Centre for Medium-range Weather Forecasts (ECMWF) with a horizontal resolution of 1/8 °. Initial and boundary conditions of the ocean model are provided by the analyses of the MERCATOR-OCEAN operational centre (Lellouche et al., 2013) based on the NEMO ocean model (Maraldi et al., 2013). These ocean fields are corrected according to the method proposed by Estournel et al. (2016a). Practically, the vertical background stratification is corrected using the CTD profiles of the MOOSE 2012





oceanographic cruise. Estournel et al. (2016a) show that this correction is necessary to properly simulate the preconditioning phase and the triggering of the convective phase. In this study, to also improve the sea surface temperature initial state, the surface temperature is restored toward the SST satellite data (Buongiorno Nardelli et al., 2013) during the 14 days preceding the beginning of the simulations using a restoring time scale of 3 days. At the end of this process, we obtain a surface temperature

that closely matches the satellite observation of 16 August. 2012. Sensitivity to the initial state is not discussed here as it has been the subject of a thorough study by Estournel et al. (2016a). During the rest of the simulation, the coupled model evolves freely, without further data assimilation and without any restoring or nudging procedure.

## 5 Results

### 5.1 Air-sea exchanges at Lion meteorological buoy

#### 5.1.1 Atmospheric and oceanic surface parameters

A first evaluation of the simulations is carried out by comparing the computed fields with the corresponding measurements at the Lion buoy. This buoy has already been used in various previous studies to validate atmospheric and/or oceanic simulations (Lebeaupin-Brossier et al., 2014; Léger et al., 2016; Rainaud et al., 2016). Figure 4 shows the temporal evolution of surface atmospheric and oceanic parameters (10 m wind speed, 2 m temperature and humidity and sea surface temperature) computed

at the buoy, together with the corresponding observations. The same surface parameters are also presented in Fig. 5 in the form of scatter plots (simulations versus observations) while the associated statistics (bias, root mean square error and correlation coefficient) are given in Table 2.

Figure 4 a allows the strong Tramontane and Mistral events to be identified (with hourly wind speeds exceeding 15m/s outlined in grey), alternating with calm wind situations. The Mistral and Tramontane episodes are systematically accompanied

by marked drops in temperature and moisture (Figs 4 b and 4 c). By a cumulative effect, the succession of strong wind events in autumn leads to a progressive decrease of the sea surface temperature (Fig. 4 d), which reaches its minimum value of 12.9 °C (i.e. the temperature of the deep water) in early winter and then remains nearly constant during the convective period.

All three simulations accurately reproduce the time evolution of the wind speed at the buoy throughout the 8-month period, with a correlation of 0.9 and a bias lower than $0.2 \, \mathrm{m \, s^{-1}}$. In particular, the timing of the strong wind events is well captured.

Moreover, the wind speed maxima are well represented (Fig. 5 a), which is essential to correctly reproduce the intense air-sea exchanges associated with convection (Herrmann and Somot, 2008). Finally, there is no significant difference in wind speed between the different simulations (Figs 4 a, 5 a and Table 2).

For the other surface atmospheric parameters (2 m air temperature and relative humidity), larger discrepancies are found from one simulation to another. Air temperature and humidity remain relatively close to observations in terms of correlation

(respectively 0.98 and 0.85, Table 2). In contrast, bias and root mean square error exhibit larger differences between simulations, especially for humidity. In particular, it is clear from Fig. 4 c that the moisture drops associated with the strong wind episodes are more pronounced in COARE and ANDREAS than in MOON.





The calculated sea surface temperature is remarkably well correlated with the observations for all simulations ($> 0.98$, Table 2). This is partly due to the correct representation of the drop of SST during the events of Tramontane and Mistral in autumn but is also influenced by the very weak variability of the SST during the winter period when the SST ceases to evolve. However, there are significant differences between simulations in autumn. In particular, the COARE SSTs appear overestimated (Figs 4 d and 5 d). During this period, the Tramontane and Mistral events produce a cooling in response to the enhanced turbulent heat fluxes on the one hand and to the temperature advection associated with the northward displacement of the NBF on the other hand (Estournel et al., 2016a). In such conditions, it is clear that a reasoning limited to the local vertical exchanges is insufficient to provide a sound interpretation of the results. Nevertheless, it can be concluded from Figs 4 and 5 and Table 2 that the results of the MOON simulation appear to agree with the Lion buoy observations better than the results of the other two simulations do.

### 5.1.2 Air-sea fluxes

The time evolution of the turbulent fluxes computed at the buoy is shown in Fig. 6. Unfortunately, as there was no flux measurement at the buoy, this figure is limited to model-model comparison. During the strong wind episodes, all the turbulent fluxes are strongly enhanced and the air-sea exchanges are intensified. As suggested by Fig. 2, the wind stress is very similar for all three simulations (Fig. 6 a), while the sensible and latent heat fluxes (Figs 6 b and c) can differ significantly from one simulation to another, especially during the strong Mistral and Tramontane wind events. During these events, in accordance with Fig. 2, the strongest sensible and latent heat fluxes are obtained with MOON and the weakest with COARE. ANDREAS produces a sensible heat flux similar to that of MOON, and a weaker latent heat flux more similar to that of COARE. For example, on 28 November (corresponding to one of the strongest wind episodes), the daily average latent heat flux reaches $1100 \, \mathrm{W \, m^{-2}}$ in MOON versus only $780 \, \mathrm{W \, m^{-2}}$ in COARE and ANDREAS. On the same day, the sensible heat flux is 390 $\mathrm{W \, m^{-2}}$ in MOON and ANDREAS compared to only $300 \, \mathrm{W \, m^{-2}}$ in COARE. In air-sea coupled simulations, the interactive evolution of ocean and atmosphere influences the turbulent heat fluxes, which themselves modify the atmospheric and oceanic surface fields (Fig. 4) involved in the flux calculation. In statically unstable Mistral and Tramontane conditions, if the sensible (or latent) heat flux increases, the vertical temperature (or humidity) gradient is reduced, which in turn limits the increase in the sensible (latent) heat flux. These feedback effects tend to reduce the differences between the simulations.

Furthermore, during the autumn, as shown in Fig. 4 d, the sea surface temperature evolves differently in each simulation. This is particularly the case for the COARE simulation, which presents significantly warmer SSTs than the other two simulations and than observations. This results in larger turbulent heat fluxes during low wind periods in autumn for the COARE simulation than for ANDREAS and MOON (Fig. 6).

To complement the analysis, the radiative fluxes and precipitation, which also contribute to the air/sea exchanges, have been analysed. As neither any major departures from the observations nor any significant differences between the simulations were found, they are not shown.



In summary, all the simulations generally give good agreement with the surface parameters observed at the Lion buoy throughout the eight months considered, although significant punctual differences may appear between the different simulations, especially during the Tramontane and Mistral events. These differences mainly affect the heat and moisture exchanges whereas the momentum exchanges are very weakly impacted. As reflected by the statistical analysis, in our coupled system, the MOON parameterization gives the best agreement with the available observations. However, considering the impossibility of directly validating the air-sea fluxes and also the multiple sources of uncertainty in such a complex modelling system (and their possible compensations) it is not clear whether the MOON flux parameterization is better per se or whether it is simply is the most suitable parameterization for our modelling system.

## 5.2 Impact of the air-sea exchanges on the oceanic stratification

### 5.2.1 Buoyancy Mass fluxes and oceanic stratification

The air-sea exchanges are now assessed through the Buoyancy Mass Flux (BMF). This flux, directly linked to turbulent fluxes but also to radiative fluxes and precipitation, is formulated as follows:

$$BMF = \alpha \frac{Q_{net}}{C_p} + \beta SSS \rho_0 (E - P) \tag{5}$$

*where $\alpha$ is the thermal expansion coefficient in $\mathrm{K}^{-1}$, $Q_{net}$ is the net heat flux (sum of the net radiative flux, and turbulent heat fluxes) in $\mathrm{W\,m}^{-2}$, $C_p$ is the specific heat capacity in $\mathrm{J\,kg}^{-1}\,\mathrm{K}^{-1}$, is the saline contraction coefficient in $\mathrm{psu}^{-1}$, SSS is the surface salinity in $\mathrm{psu}$, $\rho_0$ is a reference density in $\mathrm{kg\,m}^{-3}$, E is the evaporation and P the precipitations in $\mathrm{m\,s}^{-1}$. BMF is expressed in $\mathrm{kg\,m}^{-2}\,\mathrm{s}^{-1}$.*

The time evolution of the buoyancy mass flux computed at the Lion buoy and for each simulation is shown in Fig. 7 a (instantaneous values) and 7 b (cumulative values). Its evolution closely follows (with opposite sign) the evolution of the turbulent fluxes. The maximum buoyancy losses appear during Tramontane and Mistral events, which account for about 70 % of the total buoyancy loss. The simulations can be segmented into three periods; first the preconditioning period (16 August to 15 January) with a decrease in the cumulated buoyancy mass flux and in SST, then the convective period (15 January to 21 March) with a decrease in the cumulated buoyancy mass flux without decrease in SST and, finally, the start of the restratification period (21 March to 30 April) when the buoyancy mass flux starts to increase. As can be seen in Fig. 7 b, the buoyancy loss during the preconditioning period is nearly equivalent to the buoyancy loss during the convective period. The largest cumulative buoyancy loss at the end of the simulations is obtained with MOON (195 $\mathrm{kg\,m}^{-2}$) followed by COARE (170 $\mathrm{kg\,m}^{-2}$) and ANDREAS (165 $\mathrm{kg\,m}^{-2}$). By the end of the simulations, COARE has produced a slightly larger loss of buoyancy than ANDREAS while turbulent fluxes for the Tramontane and Mistral events are greater in ANDREAS. This is due to the larger buoyancy mass flux during the calm wind periods of the preconditioning phase in COARE.





Figure 7 c shows the time evolution of the stratification index (Eq. 4) relative to the 2000 m depth at the location of the buoy. At the beginning of the simulations the stratification is $160 \, \mathrm{kg \, m^{-2}}$, i.e. less than the total buoyancy loss produced by any of the simulations, suggesting, that in all three simulations, the water column experienced full mixing down to a depth of 2000 m at the buoy location. The decrease of the stratification index with time is not continuous and its evolution is not strictly directly

correlated with the evolution the cumulated buoyancy mass flux (as it would be in a one-dimensional system). Even if most of the stratification losses occurs mainly during the Tramontane and Mistral, during the weak wind periods, despite the slow decrease of cumulated buoyancy mass flux, the stratification index may increase.

Figure 8 places these local results in a wider context and shows the spatial distribution of the cumulated buoyancy mass fluxes during the whole simulation period , the preconditioning period, and the convective period for each simulation. During

the whole simulation period, three distinct maxima of buoyancy loss appear, two in the Gulf of Lion located within Tramontane and Mistral corridors, and a third one located further south. Whereas the first two stem from local wind maxima, the southern maximum, only present during the preconditioning period, is more related to the warm waters brought over by the seasonal northward displacement of the NBF. When the Tramontane or Mistral blows over this warm patch, the event is associated with strong air-sea temperature gradients and enhanced turbulent heat fluxes. This explains the very strong buoyancy mass flux losses

observed during the autumnal Tramontane and Mistral events at the buoy location (Fig. 7 a), which is situated in the vicinity of the frontal zone. This suggests that, during the preconditioning period, the dynamics of the NBF plays a major role in the loss of surface buoyancy in the deep water zone. Furthermore, the front displacement is modulated by the wind intensity. The increase (decrease) of stratification during the period of weak (strong) winds (Fig. 7 c) is due to the lateral advection of light (heavy) water by the northward (southward) displacement of NBF (Estournel et al., 2016a). Because of these horizontal processes and

their feedback on the buoyancy mass flux, the three simulations experience different time evolutions. The large discrepancy seen between the COARE and MOON buoyancy mass fluxes and stratification indices may also result from differences in the NBF progression. In winter, during the convective phase, only the two maxima of the Gulf of Lion remain. In the deep water zone, as the SST has reached a nearly-constant value, there is no significant effect of the SST structures on the buoyancy mass flux. At the Lion buoy, the buoyancy mass flux is not affected by the SST and the local increase in stratification during the

period of weak winds (Fig. 7 c) is principally due to the advection of light water into the mixing zone by baroclinic instability (Marshall and Schott, 1999).

The comparison of the three simulations highlights the impact of the surface flux parameterization. In MOON, the buoyancy mass flux losses are stronger than in COARE and affect a much wider area.

### 5.2.2 Validation of oceanic stratification with oceanographic cruises

To assess the evolution of the oceanic stratification, Stratification Indices collocated in space and time with all CTD and Argo profiles were calculated for each simulation and compared with the corresponding values deduced from the observations. Results were analysed in terms of bias (observations minus simulations). Figure 9 shows the spatial distribution of the SI (1000 m) bias obtained for each simulation and for the two DEWEX oceanographic cruises (leg 1 in winter 2013 and leg 2 in spring



2013) while Table 3 gives the values of the averaged SI bias computed in the convective area for different depths (1000, 1500 and 2000 m).

During the convective phase (winter 2013), as shown in Figs 3 c and e, an area of mixed profiles (with $SI(1000\text{m}) < 10\text{kg m}^{-2}$) induced by the oceanic deep convection is present in the centre of the basin. This area is surrounded by more

stratified waters corresponding to the NC to the North and to the NBF to the South. From Figs 9 a-c, it is clear that all the simulations present an excess of stratification in the convective area (negative bias) and an excess of mixing south of the NBF (positive bias). However, it is noteworthy that the SI (1000 m) bias is significantly reduced in the MOON simulation (Fig. 9 c). The good performance of MOON with respect to the stratification is further confirmed by Table 3: at 1500 and 2000 m, the SI bias reduction obtained with MOON is even more spectacular. During spring 2013, as seen in Figs 3 d and e, the stratification

index starts to slowly increase in the convection area. For all simulations, as in winter, the negative $SI(1000\text{m})$ bias is large to the south-east and the water column is too mixed while, in the convection area, the bias is fairly low, especially for the MOON simulation (Fig. 9 f). On average (Table 3) the bias is now positive and also greater with MOON than for ANDREAS and COARE. However, this is mainly due to the fact that the large positive biases present in the South of the domain in all the simulations are in ANDREAS and COARE partly compensated by large negative biases found in the centre of the convective

area (Figs 9 d-f).

In summary, the evolution of stratification is not simply related to the buoyancy mass flux but results from the complex interaction between buoyancy mass fluxes and advective processes. In this study, the MOON simulation significantly reduces the negative bias of the stratification, in the convection area, during the convective and restratification periods, again indirectly suggesting that the air-sea fluxes are most realistic in the MOON simulation.

## 5.3   Impact on the air-sea exchanges on the deep water formation

In this section observations provided by the Lion buoy and the DEWEX cruises are compared with the results of the different simulations in terms of mixed patch, timing of convective process and volume of deep dense water formed.

### 5.3.1   Mixed patch

The spatial distribution of the convection is first examined. Figure 10 shows the position of the stratified and mixed CTD and

Argo profiles and the extent of the mixed patch calculated by the model for the different simulations. The mixed patch is defined here as the grid points where the stratification index relative to 1000 m (or 2000 m) reaches zero at least once during the winter. In other words, this corresponds to the area where deep convection reaches the 1000 m (or 2000 m) depth. The CTD and Argo profiles indicate that deep convection occurs between 41 °N and 43 °N, and between 3.5 °E and 6.5 °E. Obviously the CTD and Argo profiles only show a partial view of the mixed patch as they are punctual observations, whereas the simulated mixed

patch corresponds to the area where deep convection occurred at least once during winter. The deep convection zone at 1000 m clearly appears in all the simulations but its eastern (and to a lesser extent southern) extension varies. The COARE and MOON simulations lead to the smallest and the largest mixed patch, respectively. However the observed profiles do not allow us to





conclude on the most realistic extension at 1000 m. The deep convection zone at 2000 m is much smaller than at 1000 m but for this depth, observations clearly indicate that the extent of the mixed patch is better depicted by MOON than by COARE or ANDREAS.

### 5.3.2 Timing of convective process

The development of deep convection is examined using the observations produced by the LION mooring positioned in the centre of the deep water zone. Figure 11 shows the time evolution of the observed and simulated sea water density at this point. During the preconditioning period, the surface density anomaly is less than $29.0\ \mathrm{kg\,m^{-3}}$ and the bottom density anomaly is $29.11\ \mathrm{kg\,m^{-3}}$. In the observations, the mixing reaches the first upper level of the LION mooring (150 m) in mid-January and the first complete mixing of the water column is achieved at the beginning of February. All the simulations produce deep

convection but with different timing and intensity. In the COARE and ANDREAS simulations, complete mixing occurs too late (after the second event of Mistral and Tramontane of February) while for the MOON simulation, the process occurs too early (after the second event of Mistral and Tramontane of January). After the complete mixing, a sequence of Tramontane and Mistral events led to the densification of the water column (to a density between 29.12 and $29.13\ \mathrm{kg\,m^{-3}}$). This densification in February is represented by all simulations. However, it is too weak in COARE and ANDREAS. Finally, at the beginning

of March, a period of weak wind allows surface restratification. The last event of strong Tramontane and Mistral, around 15 March, destroys this restratification and again leads to the full mixing of the water column and to an increase of the seawater density up to $29.12\ \mathrm{kg\,m^{-3}}$. This mixing event is not well represented by COARE and ANDREAS. Overall, despite the too-early full mixing, the MOON simulation gives the best representation of the deep convection at the LION mooring and correctly captures its three densification events.

### 5.3.3 Volume of deep water

The volume of the water mass created by deep convection is evaluated using the method proposed by Waldman et al. (2016b) for the same area and same period. Figure 12 shows the time evolution of the dense water formation rate computed in the simulations and estimated by Waldman et al. (2016b) with its error bar. During the autumn, the volume decreases slowly, whatever the simulation. This decrease is due to the dense water advection outside the study zone. Then, in winter, it increases

rapidly, especially during the Tramontane and Mistral events, with the development of deep convection. After the convective events, the dense water volume decreases again, due to the restratification and export processes. The timing of deep water formation and the volume created are very different according to the simulations. The timing of deep water formation is the same as that discussed for the LION mooring. The deep water volume created depends on the timing, spatial extent and intensity of deep convection processes. The COARE, ANDREAS and MOON simulations produce respectively $0.3 \times 10^{13}\ \mathrm{m^3}$

, $1.5 \times 10^{13}\ \mathrm{m^3}$ and $3.4 \times 10^{13}\ \mathrm{m^3}$ between winter 2012 and spring 2013. For the three simulations, the amount of dense water formed is lower than the one calculated from the observations $(4.5(\pm 1.1) \times 10^{13}\ \mathrm{m^3})$. However, the most realistic volume is obtained with MOON. The differences between the simulations highlight the great sensitivity of deep water formation to turbulent flux parameterization.





## 6 Conclusions

This study focused on assessing the ability of a regional ocean atmosphere coupled system, based on the SYMPHONIE, SURFEX and MESONH models, to correctly represent ocean convection and deep water formation in the NWMS. Several realistic simulations, were carried out over a period of 8 months, from summer 2012 to spring 2013, and were used to investigate
the sensitivity of the system to the parameterization of turbulent fluxes.

First, this study shows the ability of the air-sea coupled system to reproduce the evolution of the ocean and the atmosphere for several months by relying only on realistic initial and boundary conditions and without resorting to data assimilation or nudging. For all simulations, a good correlation is obtained between the observed and computed surface parameters at the Lion buoy. During the Tramontane and Mistral events, the turbulent heat fluxes differ significantly from one simulation to another,
directly impacting the atmospheric and oceanic surface parameters. In a previous study devoted to the same case study, Rainaud et al. (2016) underlined the difficulty of reproducing air surface temperature and moisture during the Mistral and Tramontane events and advocated the use of an air-sea coupled model and a purposely adjusted turbulent flux parameterization. Our study demonstrates that are strongly sensitive to the turbulent flux parameterizations, not only air surface temperature and moisture but also sea surface temperature. Our results also suggest that coupling plays a key role during the autumn storms when the
rapid drops of SST reduce the turbulent heat fluxes. In winter, the impact of the coupling is likely to be weaker since the SST does not vary much anymore. Among the three parameterizations, we found that MOON (i.e. the parameterization yielding the strongest heat turbulent fluxes) significantly reduced the bias between the observed and computed surface parameters. Unfortunately, due to the lack of flux measurements at the buoy, it was not possible to validate the computed turbulent surface fluxes directly.

Then, the buoyancy mass flux was calculated and compared to the evolution of the stratification for each simulation. The stratification evolution is directly impacted by the buoyancy mass flux loss but there is no strict correlation between stratification and buoyancy mass flux. As already shown in Estournel et al. (2016a), this confirms the importance of the advective processes on the evolution of the stratification in the deep water area. Moreover, these advective processes also directly impact the surface buoyancy mass fluxes, particularly during preconditioning period, when the position and dynamics of the North Balearic Front
clearly affect these fluxes. This interaction between the buoyancy mass fluxes and the advective processes is clearly an air-sea coupled process, which deserves to be analysed in greater depth. In terms of stratification, the effect of MOON was also found to be positive, with again a general reduction of the bias between observed and computed parameters.

Finally, the timing and the spatial extent of the convection process are very sensitive to the flux parameterization. This impacts the volume of deep water created, which varies from $0.3 \times 10^{13}\,\mathrm{m^{-3}}$ (with COARE) to $3.4 \times 10^{13}\,\mathrm{m^{-3}}$ (with MOON),
i.e. by a factor of 11.3. For the same case study, Léger et al. (2016) studied the sensitivity to initial conditions and found a factor of 4.4 between two different sets of initial conditions. Our results suggest that the deep water formation process in the NWMS might be even more sensitive to atmospheric forcing. Here again, in spite of convection being triggered too early and the volume of dense water formed being slightly underestimated, MOON appears to give the most satisfactory results.

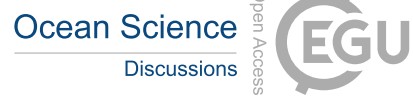



However, this conclusion regarding the good performance of the MOON flux parameterization needs to be further consolidated. Future work will investigate the sensitivity of the present results to the resolution of the atmospheric model. In the present configuration (10 and 1 km grid-spacing for the atmosphere and ocean, respectively), the local maxima and horizontal gradients of the surface atmospheric parameters are probably too smooth, which may affect the representation of the oceanic

front, known to play a major role in air-sea coupling (Small et al., 2008). A second point of interest, will be to disentangle the effect of the flux formulation from the effect of the air-sea coupling and to check whether the MOON parameterization still improves the results in the absence of coupling. A further step will be to couple the current system with a wave model and to re-visit the results obtained with the ANDREAS parameterization, which takes the significant wave height into account to calculate the depth of the sea-spray layer (Andreas et al., 1995). In the configuration used in this study, the wave height was

only very roughly estimated as a function of wind (Andreas and Wang, 2007). Finally, the good overall performance of the newly developed coupled system encourages us to tackle future studies that necessitate an accurate description of the air/sea interface such as the study of the NBF dynamics during stormy events or the Mediterranean tropical-like cyclones (Miglietta et al., 2013).

*Acknowledgements.* This work is a contribution to the MISTRALS/HyMeX programme through the ASICS-MED (ANR-12-BS06-0003)

project funded by the French National Agency for Research (ANR). Data were obtained from the HyMeX programme, sponsored by grants from MISTRALS/HyMeX and Météo-France. The authors acknowledge the international ARGO programme, the LEFE/GMMC programme and the French NAOS project for supporting the deployment of profilers. Argo and CTD data were collected and made freely available by the CORIOLIS project (http://www.coriolis.eu.org) and programmes that contribute to it. We acknowledge the crews of R/V Suroit and Tethys II and the scientists involved in the different cruises mentioned in this paper. Numerical simulations were performed using HPC resources from

CALMIP (CALcul en MIdi-Pyrénées, projects 1247, 09115 and 1325) and GENCI (Grand Equipement National de Calcul Intensif, project 010569)



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





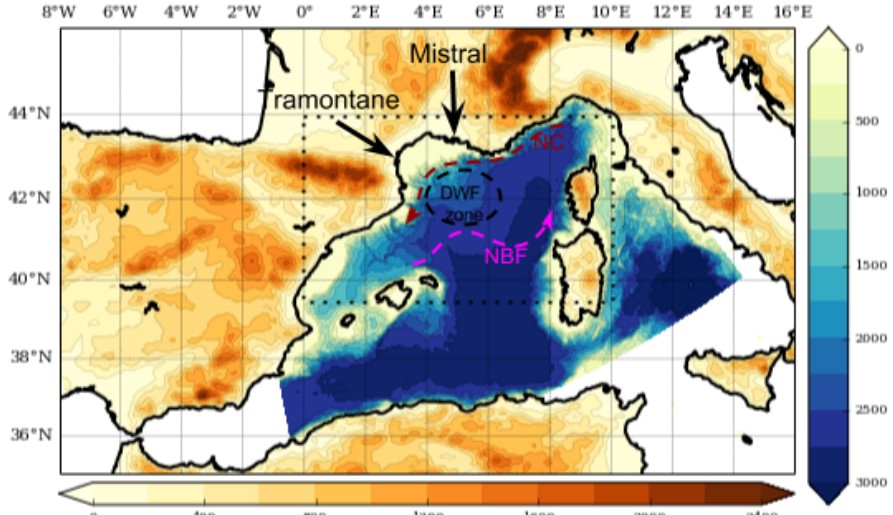

**Figure 1.** Full frame: spatial extension of the atmospheric model. Topography (meters, horizontal colour bar, positive above sea level) and bathymetry (metres, vertical colour bar, negative below sea level) of the ocean model. The black dotted rectangle indicates the area studied.

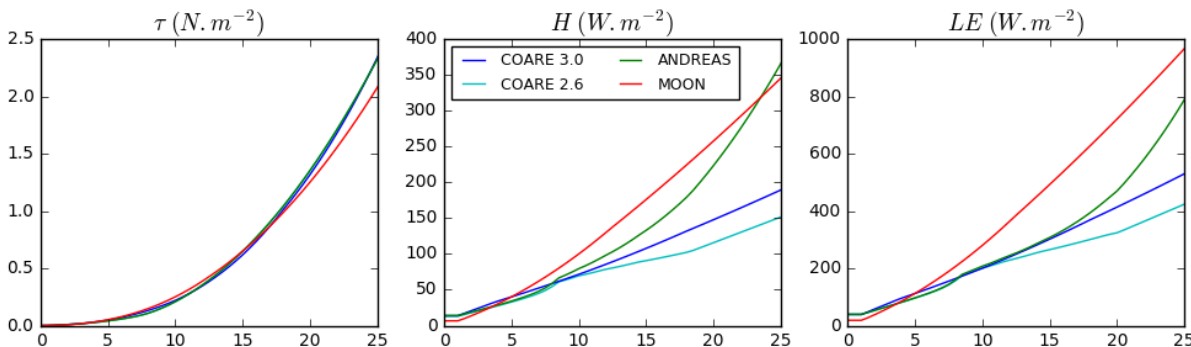

**Figure 2.** Turbulent fluxes as functions of the surface wind speed for each bulk parameterization (COARE 3.0 in blue, COARE 2.6 in cyan, ANDREAS in green and MOON in red). The computation was performed with SURFEX offline, with the air surface temperature fixed at 10 $°C$, the sea surface temperature at 15 $°C$, the relative humidity at 70 %, and the surface atmospheric pressure at 1013 hPa.

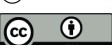



**Figure 3.** Stratification Index at 1000 m derived from CTD (squares) and ARGO (dots). Observations during (a) summer 2012 (MOOSE), (b) autumn 2012 (DOWEX), (c) winter 2013 (DEWEX Leg 1) and (d) spring 2013 (DEWEX leg 2). (e) Time series of Stratification Index at 1000 m. The red contour corresponds to the convective area defined in section 3.



**Figure 4.** Time series of (a) the 10 m wind speed (b) the 2 m air temperature (c) the 2 m air humidity and (d) the sea surface temperature measured at the Lion meteorological buoy (in black) and computed by each simulation (COARE in blue, ANDREAS in green and MOON in red).



**Figure 5.** Scatter plots of simulations against observations at Lion buoy for (a) the 10 m wind speed, (b) 2 m air temperature and (c) humidity, and (d) sea surface temperature.





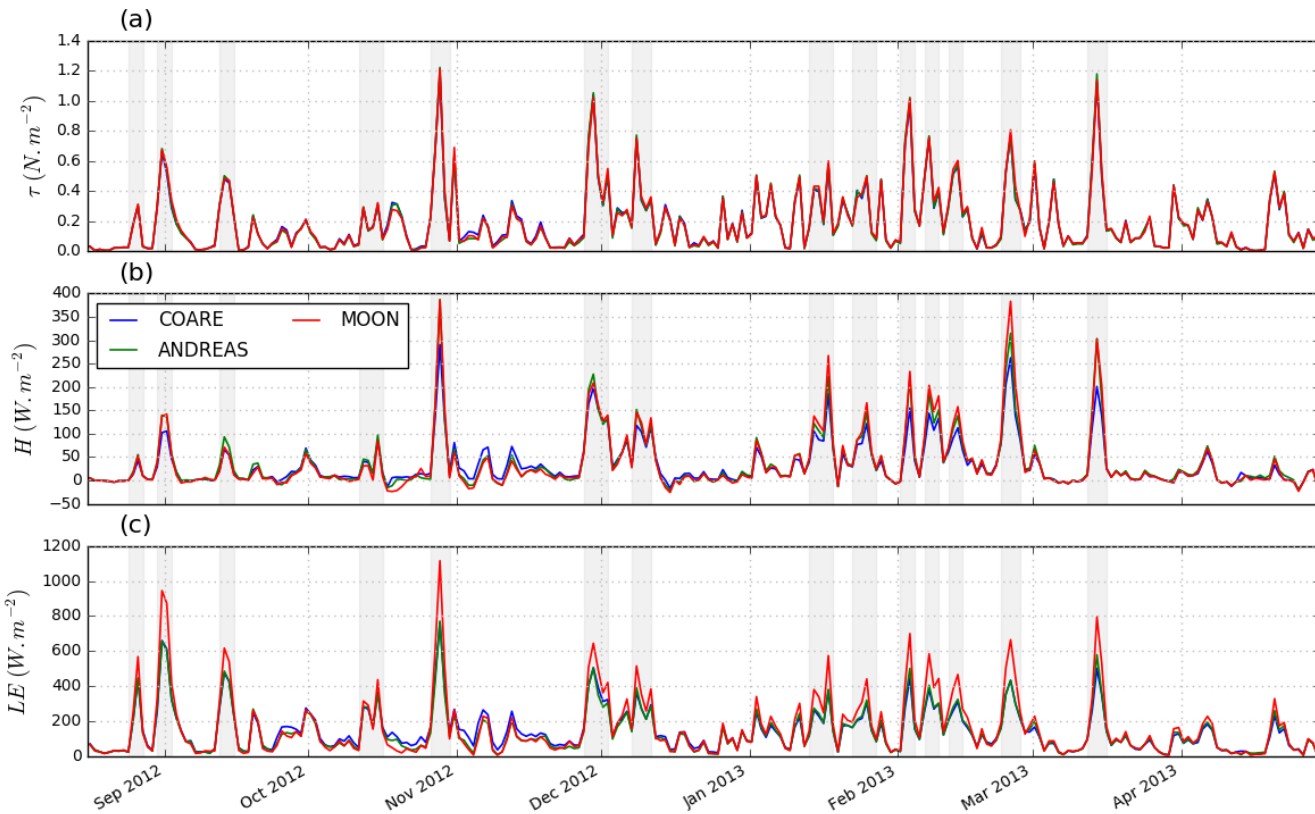

**Figure 6.** Time series of computed (a) wind stress, (b) sensible heat flux and (c) latent heat flux at the location of Lion buoy for each simulation (COARE in blue, ANDREAS in green and MOON in red).





**Figure 7.** Time series of (a) instantaneous buoyancy mass fluxes, (b) cumulated buoyancy mass fluxes (Eq. 5), and (c) stratification index (Eq. 4) relative to 2000 m at Lion buoy computed for each simulation (COARE in blue, ANDREAS in green and MOON in red).





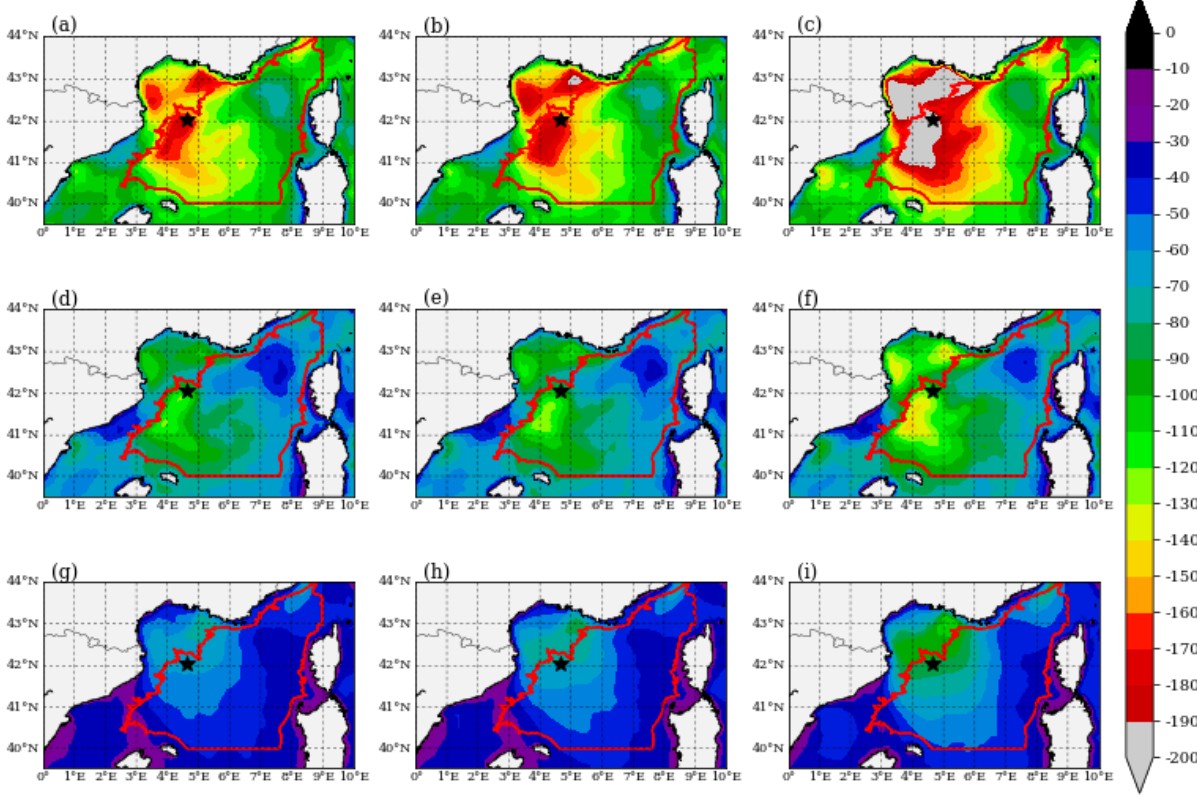

**Figure 8.** Buoyancy mass fluxes (in $\mathrm{kg\,m^{-2}}$): (a-c) during preconditioning and convective period (16 August 2012 to 21 March), (d-f) only during preconditioning period (16 August 2012 to 15 January 2013), (g-i) only during convective period (15 January 2013 to 21 March 2013) computed for each simulation: (left) COARE, (middle) ANDREAS, and (right) MOON. Red line: Convective area defined in section 3. Black star: location of Lion meteorological buoy.





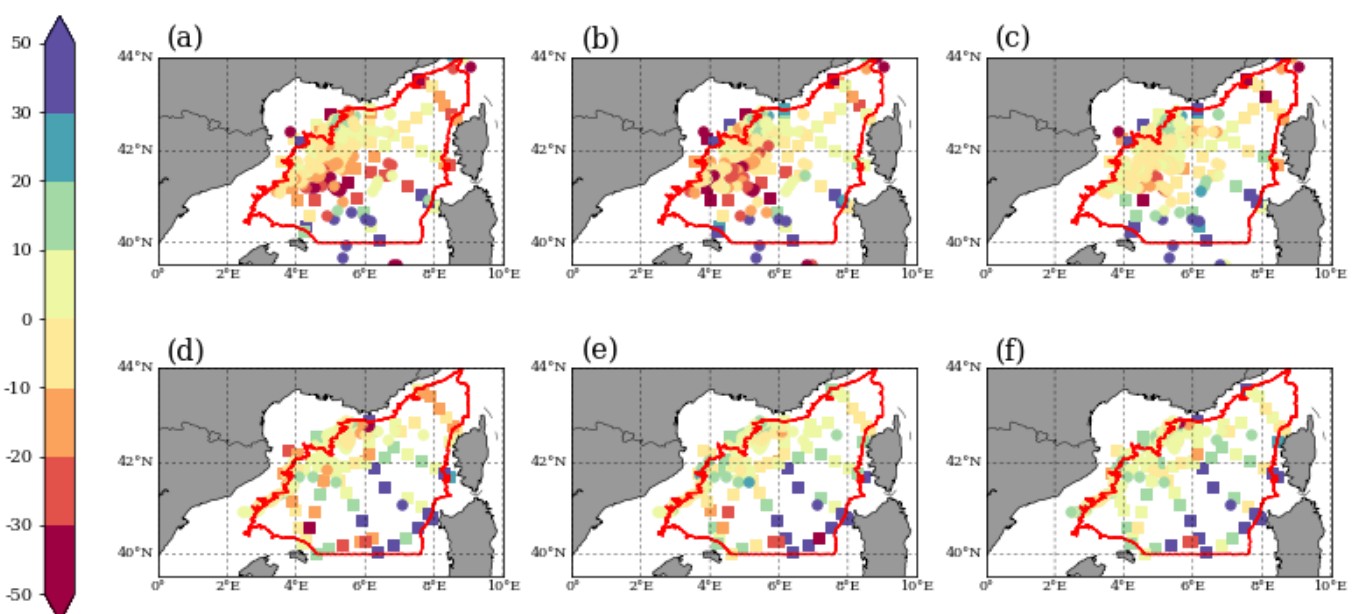

**Figure 9.** Bias of the stratification index relative to 1000 m calculated for each simulation (left) COARE, (middle) ANDREAS, (right) MOON, during (a-c) DEWEX leg 1 cruise and (d-f) DEWEX leg 2 cruise. Red line: Convective area defined in section 3.



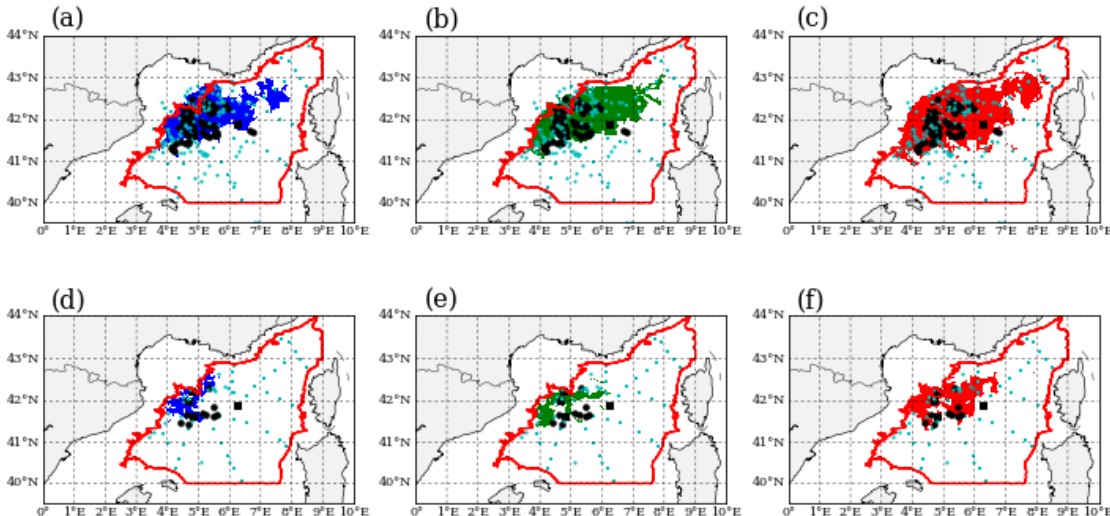

**Figure 10.** Coloured areas: extend of mixed patch zone computed for each simulation: (left) COARE, (middle) ANDREAS, (right) MOON. The mixed patch is defined as the area where the stratification index, (a-c) at 1000 m and (d-f) at 2000 m, reaches 0 during simulation. Black squares correspond to the position of the CTD and ARGO mixed profiles: (a-c) $SI(1000) = 0.$ and (d-f) $SI(2000) = 0..$ Cyan points show the position of the CTD and ARGO stratified profiles: (a-c) $SI(1000) > 0.$ and (d-f) $SI(2000) > 0.$ Red line: Convective area defined in section 3.



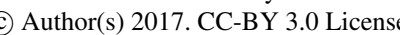



**Figure 11.** Time series of 0–2000 m sea water density (a) observed at LION mooring and computed for each simulation (b) COARE, (c) ANDREAS, (d) MOON at LION mooring location.





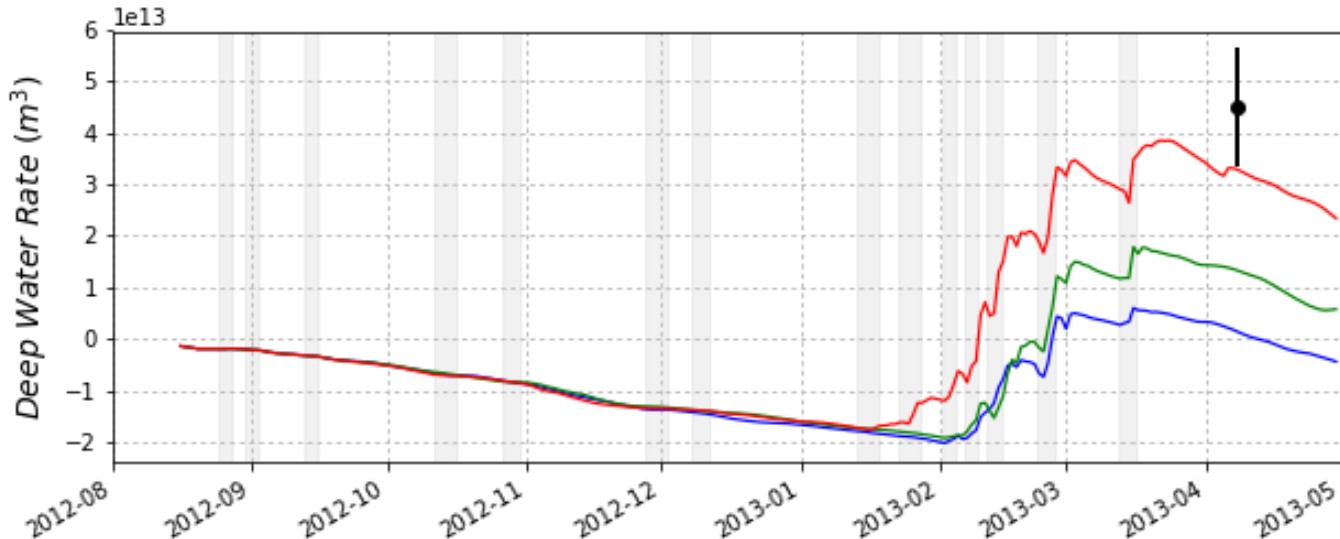

**Figure 12.** Time series of dense water volume (m$^3$) formed from 8/1/12 for sea water density up to 29.11 $\mathrm{kg\,m^{-3}}$ computed for each simulation (COARE in blue, ANDREAS in green and MOON in red) in the convection area (as defined in section 3). The black dot and error bars correspond to the dense water formation volume estimated from the observations by Waldman et al. (2016b) for the same density thresholds.





**Table 1.** Main characteristics of the atmospheric and oceanic model configurations

| Physical processes | Schemes | References |
|---|---|---|
| **Atmospheric model** | | |
| Horizontal grid | Arakava-C grid, 10 km resolution | |
| Vertical levels | Signa-z coordinates, 52 vertical levels from 15m to 15000 m | |
| Turbulence | TKE, 1D vertical | (Cuxart et al., 2000) |
| Convection | mass-flux | (Bechtold et al., 2001), (Pergaud et al., 2009) |
| Radiation | longwave : RRTM shortwave : ECMWF | (Mlawer et al., 1997), (Fouquart and Bonnel, 1980) |
| Microphysics | ICE3 | (Caniaux et al., 1994), (Pinty and Jabouille, 1998) |
| Initial and boundary condition | ECMWF analyses | |
| **Oceanic model** | | |
| Horizontal grid | Curvilinear Arakava-C grid, 1km resolution, | (Bentsen et al., 1999) |
| Vertical levels | 40 generalized sigma vertical levels | (Ulses et al., 2008) |
| Sea surface conditions | Craig & Banner TKE boundary conditions | (Estournel et al., 2009) |
| Mixing | Eddy Kinetic Energy | (Gaspar et al., 1990) |
| Convection | penetrative convection | (Deardorff et al., 1969), (Estournel et al., 2016a) |
| River input | Lateral condition (15 river inputs) | (Estournel et al., 2009) |
| Initial and boundary condition | MERCATOR-OCEAN MOOSE 2012-2013 | (Lellouche et al., 2013), (Estournel et al., 2016a) |





**Table 2.** Statistics of surface atmospheric and oceanic parameters at Lion meteorological buoy.

| EXP | COARE | | | ANDREAS | | | MOON | | |
|------|------|------|------|------|------|------|------|------|------|
| STAT | BIAS | RMS | R | BIAS | RMS | R | BIAS | RMS | R |
| U10M | -0.08 | 2.19 | 0.90 | -0.16 | 2.21 | 0.90 | 0.11 | 2.17 | 0.90 |
| T2M | -0.22 | 01.07 | 0.98 | -0.18 | 01.02 | 0.98 | -0.08 | 0.96 | 0.98 |
| HU2M | 3.24 | 07.02 | 0.85 | 3.45 | 7.23 | 0.85 | 1.15 | 6.17 | 0.85 |
| SST | -0.50 | 1.11 | 0.98 | -0.15 | 0.72 | 0.99 | 0.05 | 0.61 | 0.99 |

**Table 3.** Number of CTD-ARGO observed profiles and bias averaged for Stratification Index at 1000m, 1500m and 2000m for DEWEX oceanographic cruises.

| | Number of profiles CTD-ARGO | COARE | ANDREAS | MOON |
|------|------|------|------|------|
| DEWEX leg 1 | | | | |
| 1000 m | 62-230 | -4.9 | -3.8 | 1.8 |
| 1500 m | 56-202 | -7.4 | -5.7 | -0.4 |
| 2000 m | 48-29 | -9.2 | -6.0 | 0.9 |
| DEWEX leg 2 | | | | |
| 1000 m | 78-103 | 4.2 | 6.6 | 9.1 |
| 1500 m | 72-88 | 2.9 | 6.0 | 7.4 |
| 2000 m | 60-7 | 3.6 | 4.5 | 7.4 |