# Peer review of "Modelling deep-water formation in the North-West Mediterranean Sea with a new air-sea coupled model: sensitivity to turbulent flux parameterizations"

_Ocean Science, 2017_

## Referee Comment (RC2) · Anonymous Referee #2 · 18 Jul 2017

This paper presents a new modelling system consisting in a coupled ocean-atmosphere model and show some results regarding deep-water formation events in the North Western Mediterranean. Additionally the authors run some sensitivity experiments to show the impact of the choice of flux bulk formulas. I think the paper addresses an interesting topic, is well written and the results are interesting. Therefore I recommend it for publication after some issues are addressed. I have recommended a major review because there are many small issues to address, even if none of them are critical.

In general I think that some more details should be provided in what regards the modelling system description and the different bulk formula that are used in the paper, as these are key aspects to understand the results. Another issue is that I think the results are not discussed in depth. For instance, an important question that is now present in the modelling community is what is the role of high resolution on the modelling of these type of processes. In this sense, it is not clear to me what part of the improvement brought in this modelling system is due to the high resolution and which part due to the air-sea coupling. Some discussion on this aspect would be appreciated.

Also,the atmospheric domain looks relatively small so I wonder if the good results of the atmospheric parameters aren't induced by the lateral boundary conditions. Again, what is the role of the coupling in the good quality of the results? Could one obtain similar quality using uncoupled models? Finally, you have shown that the choice of bulk formula have small impacts on the evolution of each parameter but a huge impact on the dense water volume formed (for instance). In your opinion, what should be done to improve the parameterizations? What kind of observations would help to improve them?.

Detailed comments Page3 L4-7. As the paper has an important technical component it would be good to provide more details on the platform.

P3 L7-9. Please, provide more details on what are the conclusions of those studies. Why the air-sea coupling is benefitial? What is it providing?

Introduction. I think that the interest of using a coupled system to analyse DWF should be better presented.

P3 L30. How many levels are close to the surface?.

P3 L32. The modelling of convection is of paramount importance in this paper. Thus, more details on how this is parameterized should be included.

Section 2.1. Please, give more details about SURFEX. For non-expert readers its role

in the modelling system is confusing.

P4 L5. A "3" is missed in OASIS-MCT

Section 2.2. P4. L23. The different parameterizations used are for Cd, Ch and Ce? Please, be more clear in the description of the paremeterizations and include more details. This is also a very relevant part of the paper and the reader needs to know what are the differences between the different options.

P5 L8 "They also allow the impact of the sea spray in ANDREAS to be distinguished". I don't understand this sentence. Could you please clarify the text here?

P7.L6. Please, summarize the conclusions of Estournel et al. (2016a).

P7.L28-32. I think this paragraph is too pesimistic. The agreement between different time series is very high and differences are not so large .

P8 L1-2. Conversely I think that the extremely high correlations in the SST are overoptimistic and due to the seasonal cycle.

P8.L5-L7. Can you do a rough estimate of what is the relative importance of each mechanism (local process vs advection) ?

P8.L9-10. I think ANDREAS shows at least comparable skills with respect to MOON.

P8. L23-25. I don't understand this. It looks from the figures that differences that differences between simulations are larger during the peaks. How can you deduce that the feedback mechanism is playing a significant role?

P9. L5. I agree MOON provides the best agreement, but it is just slightly better. Considering the simulation period is relatively small I think you should moderate that statement.

P13. L13. ".. demonstrates that XXX are strongly ..." . XXX - Something is missed.

Conclusions. I don't see that MOON is really outperforming the other parameterizations. For instance, for the SI on Leg-2 COADS seems to produce better results.

Figure 1. Define in the caption what is DWF and NBF.

Figure 2 . What is each subplot? What is the x-axis?

Figure 3. What are the colours in (e) ? Isn't it redundant to use them in a time-depth plot?

Figure 4. What are the grey bars in the plots?

Table 1 "sigMa"

Table 3. Include the averaged SI index obtained from observations, so the biases can be better interpreted.

---

## Author Comment (AC1) · 4 Aug 2017

**Response to Referee #1**

The paper has been revised according to the comments from the reviewers and we thank both reviewers for their very helpful comments and suggestions. Our point-by-point response is inserted in the reviewer's comments.

Reviewer's Comment: The authors are presenting a novel study assessing the ability of a regional ocean atmosphere coupled system to correctly represent ocean convection,

especially the sensitivity of the system to the parameterization of turbulent fluxes. First, the authors assessed the model results through detailed comparisons with different observational datasets and show that the coupled system satisfactorily simulates the formation of deep water. After evaluating the uncertainties associated with the different turbulent fluxes parameterizations, the authors carried out several simulations based on 3 commonly used turbulent flux parameterizations. Their results highlight that the choice of the turbulent flux parameterization strongly influences the simulation of open ocean deep convection, especially in terms of volume of newly-formed deep water that can be different from one order of magnitude according to the parameterization choices. Open ocean deep convection plays a key role in the ocean circulation and the results found by the authors are important and will be certainly useful not only for the research groups working on the Mediterranean Sea but also in the North Atlantic, Nordic Seas, and/or Antarctic Seas. From my point of view, the manuscript represents an important contribution to our understanding of modeling deep water formation. If the scientific and presentation quality of this article are good in general, I have some (minor) comments for the authors:

RC: p5,l16 & p8,l31: What do you mean by "departure"?

Authors' Answer: These two sentences have been rephrased: "However, the difference between COARE3.0 and ANDREAS only occurred from wind speeds greater than 16 $m\ s^{-1}$ (as compared to 8 $m\ s^{-1}$ for the sensible heat flux)." "As the results were found in good agreement with the observations and did not reveal significant difference between the simulations, they are not presented here. "

RC: p6,l11: the "MAW" are not introduced before. What is the difference with the AW?

AA: The Atlantic Water corresponds to the recent surface water which enters the Mediterranean at the Gibraltar strait and spreads in the southern basin. Along its pathway into the Mediterranean basin, this Atlantic water is modified under the effect of surface heat fluxes and of vertical mixing in the convective regions. It becomes the

Modified Atlantic water (MAW).

Modified text in the manuscript "In summer (Fig. 3a), the most stratified water $(SI(1000\ m) > 120\ kg\ m^{-2})$ present in the south corresponds to recent Atlantic Water (AW), while the less stratified water $(SI(1000m) < 80\ kg\ m^{-2})$, confined to the north of the deep basin (above 42 $^\circ N$), corresponds to an older Atlantic water mass, which along its pathway into the Mediterranean basin has been modified under the effect of surface heat fluxes and of vertical mixing in the convective regions."

RC: P6,l27: "(ECMWF) with a horizontal resolution of $\frac{1}{8}$" is it the horizontal resolution of the grid choose to export the reanalysis, or is it the resolution of the atmospheric model ?

AA: It is the resolution of the ECMWF atmospheric analysis (16 km)

RC: P8,l8: What do you mean by "sound interpretation"?

AA: "sound interpretation" has been replaced by "rigorous analysis"

RC: p8,l10 : "two simulations do".two simulations.

AA: Seems correct with do.

RC: P9,l1 : "give good agreement" are in good agreement

AA: Done

RC: p9, l24: Maybe you could had that if the SST does'nt decrease at that time, it's because the mixed layer is deepening and SI continue to decrease.

AA: Done

RC: p10,l3 : "... the water column experienced ..." the water column could experience [ ...], in absence of horizontal advection

AA: Done

RC: p11,l12-15: Maybe you could quantify this , for example by calculating RMSE for each case and add them to the table. What about the excess of mixing outside the deep-mixing area, is it due to the ocean model or the air-sea flux? Is it more important using the MOON simulation? To answer these questions, it might be interesting to look at the bias in different sub-regions (Northern Current, Deep Mixing, NBF-South) instead of a single one.

AA: RMSEs have been included in Table 3 and the following text added in the manuscript: "The slightly better performance of MOON is confirmed by the RMSEs which are weaker for MOON than for COARE and ANDREAS, including for DEWEX Leg 2 at 1500 m and 2000 m depths."

The excess of mixing outside the deep-mixing area, for all simulations, is probably due to the inaccurate initial conditions in the Algerian basin.. In the northern part of the domain and south of the NBF, the initial conditions have been corrected following Estournel et al, 2016, but due to the lack of observations initial conditions remained unchanged over the Algerian basin. By advection, any excess of mixing in the Algerian basin will propagate and eventually contaminate the NBF-south region. This issue is currently under investigation and has to be addressed before running multi-year simulations.

The reviewer's suggestion regarding sub-domain analysis is interesting and was explored. However, for Dewex Leg 2, due to the limited number of observations, the results were found very sensitive to the definition of the sub-domains. These results are not included in the paper.

RC: P12, section 5.3.2 Are you expecting to really be able to simulate the exact timing of convective mixing as in the obs? Is the too early mixing in MOON due to too important BMF? By looking figure 11, the stratification at the end of January in the obs seem to be small. Maybe adding the MLD superimposed to the 4 different sections and a 5th sub-panel with a comparison of IS calculated from the simulations and the observation

would be clearer for the reader to appreciate the time evolution of the mixing and the difference between simulations and observations.

AA: To simulate the exact timing of deep convection is a tricky issue due to the weak stratification prevailing before the convective events (end of January). Small differences in the BMF in January can delay (or advance) the triggering of convection from one strong wind episode to the next (or previous) one. The too early mixing in MOON is probably induced by a too large BMF. However, this error on timing is only of a few days and the MOON simulation reproduces fairly well the succession of mixing events. As suggested by the reviewer, the MLD has been superimposed on the 3 simulated sections to ease the comparison. Unfortunately, the lack of observations in the surface boundary layer did not allow the provision of a similar figure for the observations.

RC: P13, l12-14: " Our study demonstrates that are strongly sensitive to the turbulent flux parameterizations, not only air surface temperature and moisture but also sea surface temperature" you should simplify this sentence. For example: In addition to air surface temperature and moisture , sea surface temperature is also strongly sensitive to the turbulent flux parameterizations.

AA: Done

RC: P13: l26: " In terms of stratification, the effect of MOON was also found to be positive, with again a general reduction of the bias between observed and computed parameters." you should simplify this sentence to make it clearer (e.g.: In terms of stratification, the use of MOON also led to a general reduction of the bias between observed and computed parameters.

AA: Done

RC: Table2: It seems that there are extra zero in front of some first digits.

AA: Corrected

RC: Figures: A lot of the figures are very low resolution (label difficult or impossible to

read) and should be of better quality before being published (increase dpi or export in pdf) -

AA: The quality of the figures has been improved. The new figures have been uploaded.

RC: Figures 4,6,7: you should add in the legend that the grey shaded areas correspond to strong wind periods. The grey areas are also very difficult/impossible to see and should be darker, or you could replace the grey shade by an horizontal line on the upper and lower part of each panel.

AA: The grey areas have been made darker and their meaning added to the caption.

RC: Figure 11: During the mixing period in mid-February The instrument at 500m and 700m seem to give a lower potential density that the upper and lower instruments ($> 29.12$). Is it a calibration issue or a colorbar effect ?

AA: Unfortunately, this is due to a calibration issue. The 500 m observations have been removed from the plot to avoid confusion.

Please also note the supplement to this comment:
https://www.ocean-sci-discuss.net/os-2017-43/os-2017-43-AC1-supplement.pdf

―――――――――――――――

---

## Author Comment (AC2) · 4 Aug 2017

**Response to Referee #2**

The paper has been revised according to the comments from the reviewers and we thank both reviewers for their very helpful comments and suggestions. Our point-by-point response is inserted in the reviewer's comments.

Reviewer's Comment: This paper presents a new modelling system consisting in a coupled ocean-atmosphere model and show some results regarding deep-water for-

mation events in the North Western Mediterranean. Additionally the authors run some sensitivity experiments to show the impact of the choice of flux bulk formulas. I think the paper addresses an interesting topic, is well written and the results are interesting. Therefore I recommend it for publication after some issues are addressed. I have recommended a major review because there are many small issues to address, even if none of them are critical.

In general I think that some more details should be provided in what regards the modelling system description and the different bulk formula that are used in the paper, as these are key aspects to understand the results.

Authors' Answer: Additional information on the coupling platform is now given in the introduction (see detailed comments below). In section 2, the differences between the 3 parameterisations are better presented and discussed (see detailed comments below).

RC: Another issue is that I think the results are not discussed in depth. For instance, an important question that is now present in the modelling community is what is the role of high resolution on the modelling of these type of processes. In this sense, it is not clear to me what part of the improvement brought in this modelling system is due to the high resolution and which part due to the air-sea coupling. Some discussion on this aspect would be appreciated.

AA: We agree that these issues are really important for the modelling community. However, the present study was not designed to study to role of the coupling and can not provide very relevant answers with that respect. A preliminary uncoupled experiment (not included in the paper) suggests that in absence of coupling the heat fluxes could be strongly overestimated during the preconditioning of deep ocean convection. These results are consistent with previous findings (eg. Lebeaupin Brossier and Drobinski, 2009; Small et al., 2012; Renault et al., 2012). However, it is not straightforward to distinguish the improvement resulting from the coupling itself (ie from its different feedbacks) from the one resulting from a more accurate atmospheric forcing or a more accurate flux parameterisation. This would require a series of carefully-designed experiments in which the current coupled system would be step-by-step downgraded into an uncoupled system till it would exactly mimic the behaviour of the atmospheric and ocean models in their stand-alone configuration. Currently, this type of work is hampered by the fact the surface fluxes are computed on the atmospheric grid and not on the oceanic grid.

Most of these arguments are now given in the conclusion: "However, this conclusion regarding the good performance of the MOON flux parameterization needs to be further consolidated.

First the present results were obtained with a coupled system. They could probably be different with uncoupled simulations. In air-sea coupled simulations, the interactive evolution of ocean and atmosphere influences the turbulent heat fluxes, which themselves modify the atmospheric and oceanic surface fields involved in the flux calculation. In statically unstable Mistral and Tramontane conditions, if the sensible (or latent) heat flux increases, the vertical temperature (or humidity) gradient is reduced, which in turn limits the increase in the sensible (latent) heat flux. It is likely that these feedback loop effects tend to limit the discrepancies induced differences between by the different parameterizations. The results of partial and preliminary uncoupled simulations (not shown) suggest that these discrepancies could be larger than in the coupled simulations. It would be therefore of great interest to disentangle the effect of the flux formulation from the effect of the air-sea coupling and to check whether the MOON parameterization still improves the results in uncoupled conditions. However, it is not straightforward to isolate the coupling effect in a clean and rigorous way. This requires a series of carefully-designed experiments in which the current coupled system is step-by-step downgraded into an uncoupled system till it exactly mimics the behavior of the atmospheric and ocean models in their stand-alone configuration. In our current system, this type of study is hampered by the fact that the surface fluxes are computed on

the atmospheric grid, ie at a coarser resolution that the one used by the ocean model.

The differences in resolution between the atmospheric and ocean models (10 and 1 km, respectively), though partly justified by scale considerations, is also a debatable question. A further development will thus investigate the sensitivity to the resolution of the atmospheric model. In the present configuration, the atmospheric model does not have the possibility of representing scales fully adjusted to that of the oceanic model. In particular, with a 10 km resolution, the local maxima and horizontal gradients of the surface parameters are probably too smooth, which may affect the air-sea interactions especially in the vicinity of the oceanic front (Small et al., 2008) and could also modify the response of the coupled system to the different parameterizations.

In addition, the role of the waves necessitates further investigation. In our study, the waves are not considered in COARE and MOON and only indirectly accounted for in ANDREAS. In ANDREAS, the depth of the spray layer is computed as a function of the significant wave height (Andreas et al., 1995). The latter is rather roughly estimated from a simplified parameterization based on wind speed (Andreas and Wang, 2007). Similar crude relationships are used in COARE3.0 for the wave height and wave period. Another envisaged development will couple the current system with a wave model (Michaud et al, 2012) and revisit the results obtained with the ANDREAS and COARE3.0 parameterizations.

RC: Also, the atmospheric domain looks relatively small so I wonder if the good results of the atmospheric parameters aren't induced by the lateral boundary conditions. Again, what is the role of the coupling in the good quality of the results? Could one obtain similar quality using uncoupled models?

AA: The size of the atmospheric domain is typical of the one used in the field of Numerical Weather Prediction. The fact that the model is forced at its lateral boundaries with an analysis (as opposed to a larger-scale forecast) certainly contributes to improve the results. However, most of the improvement (as compared to regional climate

results) is likely due to a better resolution of the topography and thus of the regional winds such as tramontana and mistral which are strongly controlled by the terrain. Further improvement is even expected from a higher resolution of the atmospheric model (on going work) since a 10 km resolution is still insufficient to accurately represent the atmospheric deep convective systems.

RC: Finally, you have shown that the choice of bulk formula have small impacts on the evolution of each parameter but a huge impact on the dense water volume formed (for instance). In your opinion, what should be done to improve the parameterizations? What kind of observations would help to improve them?

AA: The development of accurate flux parameterization in strong wind conditions is an area of research in itself. First, measurements in severe weather are difficult, often inaccurate and/or incomplete (e.g. simultaneous sea state observations are missing). Additional dedicated field campaigns together with wind-water tunnel experiments (Andreas et al. 2016) would certainly help to go one step further. Second, the physical processes taking place in the diphasic surface layer are complex and may be not fully understood yet. Third, there is still a large gap between our understanding of theses processes and our ability to represent them in numerical weather predictions models. This is why, as atmosphere-ocean modelers, we should remain particularly attentive to the most recent developments and multiply the experiments to test and evaluate new propositions, would they be fairly pragmatic and model-based (as Moon's) or more sophisticated and physically-based (as Andreas').

RC: Page3 L4-7. As the paper has an important technical component it would be good to provide more details on the platform.

AA: Added/modified text in the introduction "These issues, among others, have motivated the recent development of a new coupling platform (SURFEX OASIS3-MCT) providing better numerical tools to address the scientific and technical questions related to ocean-wave-atmosphere coupling (Voldoire et al., 2017). This coupling platform is
based on a external multi-surface model SURFEX (Masson et al. 2012) and on the OASIS3-MCT (Valcke et al., 2015) coupling interface. SURFEX computes the surface-atmosphere fluxes over four surface types (land, town, ocean and inland waters) and can be used in a stand-alone version with prescribed atmospheric forcing or embedded in an atmospheric models. The use of OASIS3-MCT allows SURFEX to be linked to various other models including ocean, atmosphere, hydrology, waves and sea-ice models. This generic coupling strategy based upon an externalized surface model ensures that the surface flux computations are done in a consistent way, independently of the models to be coupled. As illustrated in Voldoire in 2017, this strategy has greatly facilitated the coupling of the different models developed in the French community, including the coupling of the MESONH atmospheric model (Lafore et al., 1997) and the SYMPHONIE ocean model (Marsaleix et al., 2008,2009, 2012)"

RC: P3 L7-9. Please, provide more details on what are the conclusions of those studies. Why the air-sea coupling is beneficial? What is it providing? Introduction. I think that the interest of using a coupled system to analyse DWF should be better presented.

AA: Added text in the introduction: "Regarding the atmospheric forcing, the benefit of using a fully coupled system to study air-sea interactions in the numerical weather prediction models was already illustrated in previous studies based upon different air-sea coupled systems (eg Lebeaupin Brossier and Drobinski, 2009; Small et al., 2012; Renault et al., 2012). These studies have shown that coupled simulations provide a better representation of atmospheric and oceanic surface parameters compared to uncoupled simulations. In particular during strong wind events coupled simulations capture the rapid SST cooling more accurately, which makes the atmospheric boundary layer more stable and reduces the heat and moistures exchanges. It is likely that this improved representation of the atmospheric forcing could also lead to an improved representation of the deep water formation.

Besides the question related to coupling, there is still significant uncertainty as to the choice of a relevant parameterization to compute the turbulent fluxes for strong wind

conditions such as the Mistral and Tramontane. Current parameterizations have been carefully assessed and validated against large data sets. However, due to the limited number of available observations in strong wind conditions, they are known to be inaccurate for wind speeds exceeding 20 $m\ s^{-1}$ (e.g. Hauser et al, 2003). The sensitivity tests performed by Estournel et al., 2016b, suggest that the uncertainty associated with the turbulent flux computations could have a strong impact on the deep water formation process in the NWMS."

RC: P3 L30. How many levels are close to the surface?.

AA: 52 terrain-following vertical levels stretched from 15 m to 15000m, with 16 of them in the first km. This information is now given in the text.

RC: P3 L32. The modelling of convection is of paramount importance in this paper. Thus, more details on how this is parameterized should be included.

AA: Added text "In the case of the ocean, the vertical diffusion is parameterized following Gaspar et al. (1990) with a prognostic equation for the turbulent kinetic energy and a diagnostic relation for the mixing and dissipation lengths. A 1-km resolution is still too coarse to explicitly resolve convective plumes, which thus need to be parameterized. Different parameterizations have been proposed (e.g., Marsland et al., 2003). The most common and basic one consists in artificially increasing the vertical diffusion coefficient in statically unstable layers (eg Waldman et al., 2017). In our case, the heat and water fluxes are linearly distributed over the whole mixed layer, the depth of which is given by the depth at which the vertical density gradient becomes negative. By doing so the first level under the surface does not support the entire amount of heat loss by itself, which prevents the development of static instabilities at the surface. Furthermore, this parameterization is consistent with the nearly linear vertical variation of the buoyancy flux in the convective layer (Deardorff et al., 1969).

RC: Section 2.1. Please, give more details about SURFEX. For non-expert readers its role in the modelling system is confusing.

AA: More details about the role of SURFEX are given in the introduction (see above)

RC: P4 L5. A "3" is missed in OASIS-MCT

AA: Corrected

RC: Section 2.2. P4. L23. The different parameterizations used are for Cd, Ch and Ce? Please, be more clear in the description of the parameterizations and include more details. This is also a very relevant part of the paper and the reader needs to know what are the differences between the different options.

[revised manuscript text omitted]

RC: P5 L8 "They also allow the impact of the sea spray in ANDREAS to be distinguished". I don't understand this sentence. Could you please clarify the text here?

AA: Text replaced by "Although not used further in the following, the results of ANDREAS without sea spray effect (ANDREAS no-spray) have been added to assess its impact"

RC: P7.L6. Please, summarize the conclusions of Estournel et al. (2016a).

AA: This comment was referring to the following sentence : "Sensitivity to the initial state is not discussed here as it has been the subject of a thorough study by Estournel et al. (2016a)." which has been now removed as the conclusion of Estournel et al 2016a regarding the sensitivity of model results to the oceanic initial state was already summarized P7.L1 ". Estournel et al. (2016a) show that this correction is necessary to properly simulate the preconditioning phase and the triggering of the convective phase.".

RC: P7.L28-32. I think this paragraph is too pessimistic. The agreement between different time series is very high and differences are not so large .

AA: Rephrased sentences: "For the other surface atmospheric parameters (2 m air temperature and relative humidity), slightly larger discrepancies are found from one simulation to another. Air temperature and humidity remain relatively close to observations in terms of correlation (respectively 0.98 and 0.85, Table 2). Bias and root mean square error exhibit larger but still weak differences between simulations. The largest difference is found for humidity. In particular, it is clear from Fig. 4 c that the moisture drops associated with the strong wind episodes are more pronounced in COARE and ANDREAS than in MOON"

RC: P8 L1-2. Conversely I think that the extremely high correlations in the SST are over optimistic and due to the seasonal cycle.

AA: Rephrased sentences: "This correlation is mainly due to the representation of the seasonal cycle and to the weak variability of the SST during the winter period when the SST ceases to evolve. The drops of SST associated with the events of Tramontane and Mistral in autumn are well captured by the three simulations."

RC: P8.L5-L7. Can you do a rough estimate of what is the relative importance of each mechanism (local process vs advection) ?

AA: A rough estimate is provided by Estournel et al 2016a. Integrated during the autumn period the advection process in mass budget represent about 40% compared to local process. This information is now given in the text.

RC: P8.L9-10. I think ANDREAS shows at least comparable skills with respect to MOON.

AA: Rephrased sentences: "Nevertheless, it can be concluded from Figs 4 and 5 and Table 2 that in general the results of the MOON and ANDREAS appear to agree with the Lion buoy better than the results of the COARE do and that MOON slightly outperforms ANDREAS ."

RC: P8. L23-25. I don't understand this. It looks from the figures that differences between simulations are larger during the peaks. How can you deduce that the feedback mechanism is playing a significant role?

AA: This discussion has been moved to the conclusion where the potential impact of coupling is now discussed in more details.

RC: P9. L5. I agree MOON provides the best agreement, but it is just slightly better. Considering the simulation period is relatively small I think you should moderate that statement.

AA: Rephrased sentence: "Although the differences remain fairly weak, as reflected by the statistical analysis, in our coupled system, the MOON parameterization gives the best agreement with the available observations"

RC: P13. L13. ".. demonstrates that XXX are strongly ..." . XXX - Something is missed.

AA: Replaced sentence: "In addition to air surface temperature and moisture, sea surface temperature is also strongly sensitive to the turbulent flux parameterizations."

RC: Conclusions. I don't see that MOON is really outperforming the other parameterizations. For instance, for the SI on Leg-2 COADS seems to produce better results.

AA: It is true that the MOON bias are not the best ones for DEWEX-Leg2. The conclusion regarding the good performance of MOON has been soften and is also now supported by the analysis of the root mean square errors (which have been added in Table 3).

RC: Figure 1. Define in the caption what is DWF and NBF.

AA: Done

RC: Figure 2 . What is each subplot? What is the x-axis?

AA: This figure has been redrawn with axis labels.

RC: Figure 3. What are the colours in (e) ? Isn't it redundant to use them in a timedepth plot?

AA: The colours in Fig. 3(e) are similar to the ones used in Figs 3a-d and are defined with the colorbar, This color information is redundant in a time-SI plot, but in our opinion helphelp the visualization.

RC: Figure 4. What are the grey bars in the plots?

AA: The grey bars correspond to the strong wind periods (hourly wind speed $>$ $15 \ m \ s^{-}1$). This information is now given in the caption.

RC: Table 1 "sigMa"

AA: Corrected

RC: Table 3. Include the averaged SI index obtained from observations, so the biases can be better interpreted.

AA: Done

Please also note the supplement to this comment:
https://www.ocean-sci-discuss.net/os-2017-43/os-2017-43-AC2-supplement.pdf

**Supplement:**

[revised manuscript text omitted]

---

## Author Response (AR1)

**Response to Referee #1**

The paper has been revised according to the comments from the reviewers and we thank both reviewers for their very helpful comments and suggestions. Our point-by-point response is inserted in the reviewer's comments.

Reviewer's Comment (1): The authors are presenting a novel study assessing the ability of a regional ocean atmosphere coupled system to correctly represent ocean convection, especially the sensitivity of the system to the parameterization of turbulent fluxes. First, the authors assessed the model results through detailed comparisons with different observational datasets and show that the coupled system satisfactorily simulates the formation of deep water. After evaluating the uncertainties associated with the different turbulent fluxes parameterizations, the authors carried out several simulations based on 3 commonly used turbulent flux parameterizations. Their results highlight that the choice of the turbulent flux parameterization strongly influences the simulation of open ocean deep convection, especially in terms of volume of newly-formed deep water that can be different from one order of magnitude according to the parameterization choices. Open ocean deep convection plays a key role in the ocean circulation and the results found by the authors are important and will be certainly useful not only for the research groups working on the Mediterranean Sea but also in the North Atlantic, Nordic Seas, and/or Antarctic Seas. From my point of view, the manuscript represents an important contribution to our understanding of modeling deep water formation. If the scientific and presentation quality of this article are good in general, I have some (minor) comments for the authors:

(1): p5,l16 & p8,l31: What do you mean by "departure"?

Authors' Answer (2): These two sentences have been rephrased.

Author's changes (3): "However, the difference between COARE3.0 and ANDREAS only occurred from wind speeds greater than 16 $m\ s^{-1}$ (as compared to 8 $m\ s^{-1}$ for the sensible heat flux)." "As the results were found in good agreement with the observations and did not reveal significant difference between the simulations, they are not presented here. "

(1): p6,l11: the "MAW" are not introduced before. What is the difference with the AW?

(2): The Atlantic Water corresponds to the recent surface water which enters the Mediterranean at the Gibraltar strait and spreads in the southern basin. Along its pathway into the Mediterranean basin, this Atlantic water is modified under the effect of surface heat fluxes and of vertical mixing in the convective regions. It becomes the Modified Atlantic water (MAW).

(3): "In summer (Fig. 3a), the most stratified water ($SI(1000\ m) > 120\ kg\ m^{-2}$) present in the south corresponds to recent Atlantic Water (AW), while the less stratified water ($SI(1000m) < 80\ kg\ m^{-2}$), confined to the north of the deep basin (above 42 °N), corresponds to an older Atlantic water mass, which along its pathway into the Mediterranean basin has been modified under the effect of surface heat fluxes and of vertical mixing in the convective regions."

(1): P6,l27: "(ECMWF) with a horizontal resolution of $\frac{1}{8}$" is it the horizontal resolution of the grid choose to export the reanalysis, or is it the resolution of the atmospheric model ?

(2): It is the resolution of the ECMWF atmospheric analysis (16 km)

(1): P8,l8: What do you mean by "sound interpretation"?

(2-3): "sound interpretation" has been replaced by "rigorous analysis"

(1): p8,l10 : "two simulations do".two simulations.

(2): Seems correct with do.

(1): P9,l1 : "give good agreement" are in good agreement

(2): Done

(1): p9, l24: Maybe you could had that if the SST does'nt decrease at that time, it's because the mixed layer is deepening and SI continue to decrease.

(2): Done

(1): p10,l3 : "... the water column experienced ..." the water column could experience [ ...], in absence of horizontal advection

(2): Done

(1): p11,l12-15: Maybe you could quantify this , for example by calculating RMSE for each case and add them to the table. What about the excess of mixing outside the deep-mixing area, is it due to the ocean model or the air-sea flux? Is it more important using the MOON simulation? To answer these questions, it might be interesting to look at the bias in different sub-regions (Northern Current, Deep Mixing, NBF-South) instead of a single one.

(2): RMSEs have been included in Table 3 and the following text added in the manuscript:

(3): "The slightly better performance of MOON is confirmed by the RMSEs which are weaker for MOON than for COARE and ANDREAS, including for DEWEX Leg 2 at 1500 m and 2000 m depths."

(2): The excess of mixing outside the deep-mixing area, for all simulations, is probably due to the inaccurate initial conditions in the Algerian basin. In the northern part of the domain and south of the NBF, the initial conditions have been corrected following Estournel et al, 2016, but due to the lack of observations initial conditions remained unchanged over the Algerian basin. By advection, any excess of mixing in the Algerian basin will propagate and eventually contaminate the NBF-south region. This issue is currently under investigation and has to be addressed before running multi-year simulations.

(2): The reviewer's suggestion regarding sub-domain analysis is interesting and was explored. However, for Dewex Leg 2, due to the limited number of observations, the results were found very sensitive to the definition of the sub-domains. These results are not included in the paper.

(1): P12, section 5.3.2 Are you expecting to really be able to simulate the exact timing of convective mixing as in the obs? Is the too early mixing in MOON due to too important BMF? By looking figure 11, the stratification at the end of January in the obs seem to be small. Maybe adding the MLD superimposed to the 4 different sections and a 5th sub-panel with a comparison of IS calculated from the simulations and the observation would be clearer for the reader to appreciate the time evolution of the mixing and the difference between simulations and observations.

(2): To simulate the exact timing of deep convection is a tricky issue due to the weak stratification prevailing before the convective events (end of January). Small differences in the BMF in January can delay (or advance) the triggering of convection from one strong wind episode to the next (or previous) one. The too early mixing in MOON is probably induced by a too large BMF. However, this error on timing is only of a few days and the MOON simulation reproduces fairly well the succession of mixing events. As suggested by the reviewer, the MLD has been superimposed on the 3 simulated sections to ease the comparison. Unfortunately, the lack of observations in the surface boundary layer did not allow the provision of a similar figure for the observations.

(1): P13, l12-14: " Our study demonstrates that are strongly sensitive to the turbulent flux parameterizations, not only air surface temperature and moisture but also sea surface temperature" you should simplify this sentence. For example: In addition to air surface temperature and moisture , sea surface temperature is also strongly sensitive to the turbulent flux parameterizations.

(2): Done

(1): P13: l26: " In terms of stratification, the effect of MOON was also found to be positive, with again a general reduction of the bias between observed and computed parameters." you should simplify this sentence to make it clearer (e.g.: In terms of stratification, the use of MOON also led to a general reduction of the bias between observed and computed parameters.

(2): Done

(1): Table2: It seems that there are extra zero in front of some first digits.

(2): Corrected

(1): Figures: A lot of the figures are very low resolution (label difficult or impossible to read) and should be of better quality before being published (increase dpi or export in pdf) -

(2): The quality of the figures has been improved. The new figures have been uploaded.

(1): Figures 4,6,7: you should add in the legend that the grey shaded areas correspond to strong wind periods. The grey areas are also very difficult/impossible to see and should be darker, or you could replace the grey shade by an horizontal line on the upper and lower part of each panel.

(2): The grey areas have been made darker and their meaning added to the caption.

5   (1): Figure 11: During the mixing period in mid-February The instrument at 500m and 700m seem to give a lower potential density that the upper and lower instruments ($> 29.12$). Is it a calibration issue or a colorbar effect ?

(2): Unfortunately, this is due to a calibration issue. The 500 m observations have been removed from the plot to avoid confusion.

**Response to Referee #2**

10   The paper has been revised according to the comments from the reviewers and we thank both reviewers for their very helpful comments and suggestions. Our point-by-point response is inserted in the reviewer's comments.

Reviewer's Comment (1): This paper presents a new modelling system consisting in a coupled ocean-atmosphere model and show some results regarding deep-water formation events in the North Western Mediterranean. Additionally the authors run some sensitivity experiments to show the impact of the choice of flux bulk formulas. I think the paper addresses an interesting

15   topic, is well written and the results are interesting. Therefore I recommend it for publication after some issues are addressed. I have recommended a major review because there are many small issues to address, even if none of them are critical.

In general I think that some more details should be provided in what regards the modelling system description and the different bulk formula that are used in the paper, as these are key aspects to understand the results.

Authors' Answer (2): Additional information on the coupling platform is now given in the introduction (see detailed com-

20   ments below). In section 2, the differences between the 3 parameterisations are better presented and discussed (see detailed comments below).

(1): Another issue is that I think the results are not discussed in depth. For instance, an important question that is now present in the modelling community is what is the role of high resolution on the modelling of these type of processes. In this sense, it is not clear to me what part of the improvement brought in this modelling system is due to the high resolution and which part

25   due to the air-sea coupling. Some discussion on this aspect would be appreciated.

(2): We agree that these issues are really important for the modelling community. However, the present study was not designed to study to role of the coupling and can not provide very relevant answers with that respect. A preliminary uncoupled experiment (not included in the paper) suggests that in absence of coupling the heat fluxes could be strongly overestimated during the preconditioning of deep ocean convection. These results are consistent with previous findings (eg. Lebeaupin Brossier

30   and Drobinski (2009);Small et al. (2012);Renault et al. (2012)). However, it is not straightforward to distinguish the improvement resulting from the coupling itself (ie from its different feedbacks) from the one resulting from a more accurate atmospheric forcing or a more accurate flux parameterisation. This would require a series of carefully-designed experiments in which the current coupled system would be step-by-step downgraded into an uncoupled system till it would exactly mimic the behaviour of the atmospheric and ocean models in their stand-alone configuration. Currently, this type of work is hampered by the fact

35   the surface fluxes are computed on the atmospheric grid and not on the oceanic grid.

Author's changes (3): Most of these arguments are now given in the conclusion:

"However, this conclusion regarding the good performance of the MOON flux parameterization needs to be further consolidated.

First the present results were obtained with a coupled system. They could probably be different with uncoupled simulations.

40   In air-sea coupled simulations, the interactive evolution of ocean and atmosphere influences the turbulent heat fluxes, which themselves modify the atmospheric and oceanic surface fields involved in the flux calculation. In statically unstable Mistral and Tramontane conditions, if the sensible (or latent) heat flux increases, the vertical temperature (or humidity) gradient is reduced, which in turn limits the increase in the sensible (latent) heat flux. It is likely that these feedback loop effects tend to limit the discrepancies induced differences between by the different parameterizations. The results of partial and preliminary uncoupled

45   simulations (not shown) suggest that these discrepancies could be larger than in the coupled simulations. It would be therefore

of great interest to disentangle the effect of the flux formulation from the effect of the air-sea coupling and to check whether the MOON parameterization still improves the results in uncoupled conditions. However, it is not straightforward to isolate the coupling effect in a clean and rigorous way. This requires a series of carefully-designed experiments in which the current coupled system is step-by-step downgraded into an uncoupled system till it exactly mimics the behavior of the atmospheric and ocean models in their stand-alone configuration. In our current system, this type of study is hampered by the fact that the surface fluxes are computed on the atmospheric grid, ie at a coarser resolution that the one used by the ocean model.

The differences in resolution between the atmospheric and ocean models (10 and 1 km, respectively), though partly justified by scale considerations, is also a debatable question. A further development will thus investigate the sensitivity to the resolution of the atmospheric model. In the present configuration, the atmospheric model does not have the possibility of representing scales fully adjusted to that of the oceanic model. In particular, with a 10 km resolution, the local maxima and horizontal gradients of the surface parameters are probably too smooth, which may affect the air-sea interactions especially in the vicinity of the oceanic front (Small et al., 2008) and could also modify the response of the coupled system to the different parameterizations.

In addition, the role of the waves necessitates further investigation. In our study, the waves are not considered in COARE and MOON and only indirectly accounted for in ANDREAS. In ANDREAS, the depth of the spray layer is computed as a function of the significant wave height (Andreas et al., 1995). The latter is rather roughly estimated from a simplified parameterization based on wind speed (Andreas and Wang, 2007). Similar crude relationships are used in COARE3.0 for the wave height and wave period. Another envisaged development will couple the current system with a wave model (Michaud et al, 2012) and revisit the results obtained with the ANDREAS and COARE3.0 parameterizations.

(1): Also, the atmospheric domain looks relatively small so I wonder if the good results of the atmospheric parameters aren't induced by the lateral boundary conditions. Again, what is the role of the coupling in the good quality of the results? Could one obtain similar quality using uncoupled models?

(2): The size of the atmospheric domain is typical of the one used in the field of Numerical Weather Prediction. The fact that the model is forced at its lateral boundaries with an analysis (as opposed to a larger-scale forecast) certainly contributes to improve the results. However, most of the improvement (as compared to regional climate results) is likely due to a better resolution of the topography and thus of the regional winds such as tramontana and mistral which are strongly controlled by the terrain. Further improvement is even expected from a higher resolution of the atmospheric model (on going work) since a 10 km resolution is still insufficient to accurately represent the atmospheric deep convective systems.

(1): Finally, you have shown that the choice of bulk formula have small impacts on the evolution of each parameter but a huge impact on the dense water volume formed (for instance). In your opinion, what should be done to improve the parameterizations? What kind of observations would help to improve them?

(2): The development of accurate flux parameterization in strong wind conditions is an area of research in itself. First, measurements in severe weather are difficult, often inaccurate and/or incomplete (e.g. simultaneous sea state observations are missing). Additional dedicated field campaigns together with wind-water tunnel experiments (Andreas et al. 2016) would certainly help to go one step further. Second, the physical processes taking place in the diphasic surface layer are complex and may be not fully understood yet. Third, there is still a large gap between our understanding of theses processes and our ability to represent them in numerical weather predictions models. This is why, as atmosphere-ocean modelers, we should remain particularly attentive to the most recent developments and multiply the experiments to test and evaluate new propositions, would they be fairly pragmatic and model-based (as Moon's) or more sophisticated and physically-based (as Andreas').

(1): Page3 L4-7. As the paper has an important technical component it would be good to provide more details on the platform.

(2): Added/modified text in the introduction

(3): "These issues, among others, have motivated the recent development of a new coupling platform (SURFEX OASIS3-MCT) providing better numerical tools to address the scientific and technical questions related to ocean-wave-atmosphere coupling (Voldoire et al., 2017). This coupling platform is based on a external multi-surface model SURFEX (Masson et al., 2012) and on the OASIS3-MCT (Valcke et al., 2015) coupling interface. SURFEX computes the surface-atmosphere fluxes over four surface types (land, town, ocean and inland waters) and can be used in a stand-alone version with prescribed atmospheric forcing or embedded in an atmospheric models. The use of OASIS3-MCT allows SURFEX to be linked to various other

models including ocean, atmosphere, hydrology, waves and sea-ice models. This generic coupling strategy based upon an externalized surface model ensures that the surface flux computations are done in a consistent way, independently of the models to be coupled. As illustrated in Voldoire et al. (2017), this strategy has greatly facilitated the coupling of the different models developed in the French community, including the coupling of the MESONH atmospheric model (Lafore et al., 1997) and the SYMPHONIE ocean model (Marsaleix et al., 2008, 2009, 2012)"

(1): P3 L7-9. Please, provide more details on what are the conclusions of those studies. Why the air-sea coupling is beneficial? What is it providing? Introduction. I think that the interest of using a coupled system to analyse DWF should be better presented.

(2): Added text in the introduction:

(3)"Regarding the atmospheric forcing, the benefit of using a fully coupled system to study air-sea interactions in the numerical weather prediction models was already illustrated in previous studies based upon different air-sea coupled systems (eg Lebeaupin Brossier and Drobinski, 2009; Small et al., 2012; Renault et al., 2012). These studies have shown that coupled simulations provide a better representation of atmospheric and oceanic surface parameters compared to uncoupled simulations. In particular during strong wind events coupled simulations capture the rapid SST cooling more accurately, which makes the atmospheric boundary layer more stable and reduces the heat and moistures exchanges. It is likely that this improved representation of the atmospheric forcing could also lead to an improved representation of the deep water formation.

Besides the question related to coupling, there is still significant uncertainty as to the choice of a relevant parameterization to compute the turbulent fluxes for strong wind conditions such as the Mistral and Tramontane. Current parameterizations have been carefully assessed and validated against large data sets. However, due to the limited number of available observations in strong wind conditions, they are known to be inaccurate for wind speeds exceeding $20 \ m \ s^{-1}$ (e.g. Hauser et al, 2003). The sensitivity tests performed by Estournel et al., 2016b, suggest that the uncertainty associated with the turbulent flux computations could have a strong impact on the deep water formation process in the NWMS."

(1): P3 L30. How many levels are close to the surface?.

(2): 52 terrain-following vertical levels stretched from 15 m to 15000m, with 16 of them in the first km. This information is now given in the text.

(1): P3 L32. The modelling of convection is of paramount importance in this paper. Thus, more details on how this is parameterized should be included.

(3): "In the case of the ocean, the vertical diffusion is parameterized following Gaspar et al. (1990) with a prognostic equation for the turbulent kinetic energy and a diagnostic relation for the mixing and dissipation lengths. A 1-km resolution is still too coarse to explicitly resolve convective plumes, which thus need to be parameterized. Different parameterizations have been proposed (e.g., Marsland et al. (2003)). The most common and basic one consists in artificially increasing the vertical diffusion coefficient in statically unstable layers (eg Waldman et al. (2016a)). In our case, the heat and water fluxes are linearly distributed over the whole mixed layer, the depth of which is given by the depth at which the vertical density gradient becomes negative. By doing so the first level under the surface does not support the entire amount of heat loss by itself, which prevents the development of static instabilities at the surface. Furthermore, this parameterization is consistent with the nearly linear vertical variation of the buoyancy flux in the convective layer (Deardorff et al., 1969).

(1): Section 2.1. Please, give more details about SURFEX. For non-expert readers its role in the modelling system is confusing.

(2): More details about the role of SURFEX are given in the introduction (see above)

(1): P4 L5. A "3" is missed in OASIS-MCT

(2): Corrected

(1): Section 2.2. P4. L23. The different parameterizations used are for Cd, Ch and Ce? Please, be more clear in the description of the parameterizations and include more details. This is also a very relevant part of the paper and the reader needs to know what are the differences between the different options.

(3): Modified text in section 2. Turbulent air fluxes at the air/sea interface are computed from bulk type parameterizations based on the Monin-Obukhov similarity theory (Foken, 2006). These parameterizations compute the turbulent fluxes as

$$|\tau| = \rho_a u^{*2}$$
$$H = -\rho_a C_p u^* \theta^*$$
5    $$LE = -\rho_a L_e u^* q^*$$

[revised manuscript text omitted]

(1): P5 L8 "They also allow the impact of the sea spray in ANDREAS to be distinguished". I don't understand this sentence. Could you please clarify the text here?

(3): Text replaced by "Although not used further in the following, the results of ANDREAS without sea spray effect (AN-
45    DREAS no-spray) have been added to assess its impact"

(1): P7.L6. Please, summarize the conclusions of Estournel et al. (2016a).

(2-3): This comment was referring to the following sentence : "Sensitivity to the initial state is not discussed here as it has been the subject of a thorough study by Estournel et al. (2016a)." which has been now removed as the conclusion of Estournel et al. (2016a) regarding the sensitivity of model results to the oceanic initial state was already summarized P7.L1 ". Estournel et al. (2016a) show that this correction is necessary to properly simulate the preconditioning phase and the triggering of the convective phase.".

(1): P7.L28-32. I think this paragraph is too pessimistic. The agreement between different time series is very high and differences are not so large.

(3): Rephrased sentences: "For the other surface atmospheric parameters (2 m air temperature and relative humidity), slightly larger discrepancies are found from one simulation to another. Air temperature and humidity remain relatively close to observations in terms of correlation (respectively 0.98 and 0.85, Table 2). Bias and root mean square error exhibit larger but still weak differences between simulations. The largest difference is found for humidity. In particular, it is clear from Fig. 4 c that the moisture drops associated with the strong wind episodes are more pronounced in COARE and ANDREAS than in MOON"

(1): P8 L1-2. Conversely I think that the extremely high correlations in the SST are over optimistic and due to the seasonal cycle.

(3): Rephrased sentences: "This correlation is mainly due to the representation of the seasonal cycle and to the weak variability of the SST during the winter period when the SST ceases to evolve. The drops of SST associated with the events of Tramontane and Mistral in autumn are well captured by the three simulations."

(1): P8.L5-L7. Can you do a rough estimate of what is the relative importance of each mechanism (local process vs advection) ?

(2): A rough estimate is provided by Estournel et al 2016a. Integrated during the autumn period the advection process in mass budget represent about 40% compared to local process. This information is now given in the text.

(1): P8.L9-10. I think ANDREAS shows at least comparable skills with respect to MOON.

(3): Rephrased sentences: "Nevertheless, it can be concluded from Figs 4 and 5 and Table 2 that in general the results of the MOON and ANDREAS appear to agree with the Lion buoy better than the results of the COARE do and that MOON slightly outperforms ANDREAS ."

(1): P8. L23-25. I don't understand this. It looks from the figures that differences between simulations are larger during the peaks. How can you deduce that the feedback mechanism is playing a significant role?

(2): This discussion has been moved to the conclusion where the potential impact of coupling is now discussed in more details.

(1): P9. L5. I agree MOON provides the best agreement, but it is just slightly better. Considering the simulation period is relatively small I think you should moderate that statement.

(3): Rephrased sentence: "Although the differences remain fairly weak, as reflected by the statistical analysis, in our coupled system, the MOON parameterization gives the best agreement with the available observations"

(1): P13. L13. ".. demonstrates that XXX are strongly ..." . XXX - Something is missed.

(3): Replaced sentence: "In addition to air surface temperature and moisture, sea surface temperature is also strongly sensitive to the turbulent flux parameterizations."

(1): Conclusions. I don't see that MOON is really outperforming the other parameterizations. For instance, for the SI on Leg-2 COADS seems to produce better results.

(2): It is true that the MOON bias are not the best ones for DEWEX-Leg2. The conclusion regarding the good performance of MOON has been soften and is also now supported by the analysis of the root mean square errors (which have been added in Table 3).

(1): Figure 1. Define in the caption what is DWF and NBF.
(2): Done

(1): Figure 2 . What is each subplot? What is the x-axis?
(2): This figure has been redrawn with axis labels.

(1): Figure 3. What are the colours in (e) ? Isn't it redundant to use them in a time-depth plot?
(2): The colours in Fig. 3(e) are similar to the ones used in Figs 3a-d and are defined with the colorbar, This color information is redundant in a time-SI plot, but in our opinion helphelp the visualization.

(1): Figure 4. What are the grey bars in the plots?
(2): The grey bars correspond to the strong wind periods (hourly wind speed $> 15\ m\ s^-1$). This information is now given in the caption.

(1): Table 1 "sigMa"
(2): Corrected

(1): Table 3. Include the averaged SI index obtained from observations, so the biases can be better interpreted.
(2): Done

**Summary of all changes made in the manuscript**

Title: Unchanged

Abstract: Unchanged

Introduction: We have added information on the benefit of using a fully coupled system and on the coupled system.

Section 2.1: We have added information on the oceanic convection parameterization.

Section 2.1: We have added information on the turbulent fluxes parameterizations.

Section 3: few rewording, mainly unchanged.

Section 4: few rewording, mainly unchanged.

Section 5: few rewording, mainly unchanged.

Conclusions : We have added discussion on air-sea coupling, atmospheric resolution and waves impact on the turbulent fluxes.

[revised manuscript text omitted]

---

## Referee Report (RR1)

Review OS: Seyfried et al. 2017

I have read through the replies from the authors to both referee, and found the revised manuscript clearer and better structured. Particularly, I am happy with the response to my comments (referee#1). I think there are no scientific issues to raise and the manuscript is ready for publication. Nevertheless there are still some typesetting issues that the authors should consider.

P3,l16: extra parenthesis
p3,l19: an atmospheric model
p4,l19-20: "In our case, the heat and water fluxes are linearly distributed over the whole mixed layer, the depth of which is given by the depth at which the vertical density gradient becomes negative." → meaning not clear to much repetition (depth, which). Try to make shorter sentences.
P5, l26: thee
p15,l14 "to limit the discrepancies induced differences between by the different parameterizations." I dont understand the meaning of this sentence. Can you reformulate?

---

## Author Response (AR2)

**Response to Referee #1**

Reviewer's Comment (1): I have read through the replies from the authors to both referee, and found the revised manuscript clearer and better structured. Particularly, I am happy with the response to my comments (referee 1). I think there are no scientific issues to raise and the manuscript is ready for publication. Nevertheless there are still some typesetting issues that the authors should consider.

(1) P3,l16: extra parenthesis
Authors' Answer (2): Done

(1) p3,l19: an atmospheric model
(2): Done

(1) p4,l19-20: "In our case, the heat and water fluxes are linearly distributed over the whole mixed layer, the depth of which is given by the depth at which the vertical density gradient becomes negative." meaning not clear to much repetition (depth, which). Try to make shorter sentences.
(2) : Rephrased sentences "In our case, the heat and water fluxes are linearly distributed over the whole mixed layer. The mixed layer is defined by the depth at which the vertical density gradient becomes negative."

(1) P5, l26: thee
(2): Done

(1) p15,l14 "to limit the discrepancies induced differences between by the different parameterizations." I dont understand the meaning of this sentence. Can you reformulate?
(2) : Rephrased sentences "It is likely that these feedback loop effects tend to limit the discrepancies between the different parameterizations."

**Summary of all changes made in the manuscript**

The technical corrections of the referee were taken into account. In the previous version we estimated a factor 11.3 between our different simulations because our reference level was in summer 2012. But looking at the Fig. 12 we think that was not a fair evaluation. Thus, to clarify the Fig. 12 description and help the comparison with other studies the volume of newly formed deep water has been calculated only during increasing phases of this volume. These new numbers have been added in part 5.3.3 and in the abstract and conclusion. We apologize for this last minute changes but we think that it is really more representative of the deep water evolution.

[revised manuscript text omitted]